# *RLVMR*: Reinforcement Learning with Verifiable Meta-Reasoning Rewards for Robust Long-Horizon Agents

Zijing Zhang[1,2,*]     Ziyang Chen[1,†]     Mingxiao Li[1]     Zhaopeng Tu[1,†]     Xiaolong Li[1]

[1]Tencent Multimodal Department, Digital Human Center.     [2]Peking University

## Abstract

The development of autonomous agents for complex, long-horizon tasks is a central goal in AI. However, dominant training paradigms face a critical limitation: reinforcement learning (RL) methods that optimize solely for final task success often reinforce flawed or inefficient reasoning paths, a problem we term **inefficient exploration**. This leads to agents that are brittle and fail to generalize, as they learn to find solutions without learning *how* to reason coherently. To address this, we introduce **RLVMR**, a novel framework that integrates dense, process-level supervision into end-to-end RL by rewarding verifiable, meta-reasoning behaviors. RLVMR equips an agent to explicitly tag its cognitive steps—such as planning, exploration, and reflection—and provides programmatic, rule-based rewards for actions that contribute to effective problem-solving. These process-centric rewards are combined with the final outcome signal and optimized using a critic-free policy gradient method. On the challenging ALFWorld and ScienceWorld benchmarks, RLVMR achieves new state-of-the-art results, with our 7B model reaching an 83.6% success rate on the most difficult unseen task split. Our analysis confirms these gains stem from improved reasoning quality, including significant reductions in redundant actions and enhanced error recovery, leading to more robust, efficient, and interpretable agents.

## 1 Introduction

The quest to build autonomous agents capable of solving complex, long-horizon tasks has gained significant momentum with the rise of Large Language Models (LLMs) (Wang et al., 2022; Zeng et al., 2024; Bai et al., 2024). However, dominant training paradigms face a fundamental trade-off. On one hand, Supervised Fine-Tuning (SFT) on expert trajectories can teach agents efficient behaviors, but these policies are often brittle and fail to generalize to novel situations (Chu et al., 2025). On the other hand, RL from environmental feedback encourages exploration and can lead to better generalization, but it typically optimizes for a single, sparse reward signal: final task success.

This reliance on outcome-only rewards raises a critical, yet underexplored question: *Are agents learning to reason coherently, or are they just finding brittle shortcuts to success?* Our work investigates a pervasive issue we term **inefficient exploration**, where agents are rewarded for successful outcomes even when their path to success is built on flawed, or redundant reasoning. This leads to agents that exhibit high rates of repetitive actions and struggle to adapt to unseen tasks, because their underlying problem-solving process is unsound. Standard RL inadvertently reinforces any successful trajectory, failing to distinguish between robust and flawed reasoning processes. This deficiency undermines agent reliability and generalization, especially as tasks grow in complexity.

We argue that to build truly robust and generalizable agents, we must move beyond rewarding only the final outcome and begin to supervise the reasoning *process* itself. Inspired by metacognitive theory (Martinez, 2006), which posits that effective problem-solving depends on "thinking about

---

*This work was done during an internship at Tencent.

†Correspondence to: Zhaopeng Tu <zptu@tencent.com> and Ziyang Chen<willzychen@tencent.com>.

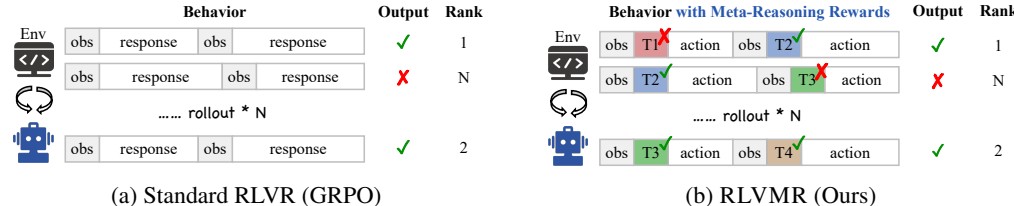

Figure 1: Comparison of LLM agent RL training paradigms: (a) Standard RL with outcome-only rewards (e.g., GRPO) inadvertently reinforces trajectories with inefficient or illogical intermediate reasoning steps. (b) Our RLVMR approach provides dense, verifiable rewards for beneficial meta-reasoning behaviors (e.g., T1-T4), directly shaping a more robust and coherent reasoning process.

thinking", we propose to directly reward beneficial cognitive behaviors. Our key insight is that high-level skills like planning, monitoring progress, exploring alternatives, and reflecting on errors can be operationalized as distinct, verifiable steps within an agent's reasoning process.

To this end, we introduce **Reinforcement Learning with Verifiable Meta-Reasoning Rewards (RLVMR)**, a novel framework that integrates dense, process-level supervision into end-to-end RL. As illustrated in Figure 1, RLVMR contrasts with standard RL by rewarding not only the final outcome but also the intermediate reasoning steps. Our framework defines a set of core meta-reasoning behaviors — *planning*, *exploration*, and *reflection/monitoring* — and enables the agent to articulate its cognitive state through special tags. During online interaction, we use lightweight, programmatic rules to grant verifiable rewards for these behaviors. For example, an 'exploration' tag is rewarded when the agent discovers a new state, while a 'reflection' tag is rewarded when it leads to the correction of a prior mistake. These process-centric rewards are combined with the global outcome reward and optimized using a policy gradient method. After a brief "cold-start" supervised fine-tuning (SFT) phase on only 200 trajectories to learn the tag syntax, the agent is trained entirely through environmental interaction.

We demonstrate the effectiveness of RLVMR on two challenging long-horizon benchmarks, ALF-World and ScienceWorld. Our experiments show that RLVMR achieves new state-of-the-art results across all settings. Notably, on the hardest unseen task split (L2), our 7B model achieves an 83.6% success rate, and surpasses the performance of the much larger models. In-depth analysis reveals that these gains are driven by a tangible improvement in reasoning quality: RLVMR-trained agents exhibit significant reductions in repetitive and invalid actions. This confirms that by rewarding the *process* of good reasoning, we create agents that are not only more successful but also more robust, efficient, and generalizable.

In summary, our contributions are as follows:

1. We identify and analyze a critical inefficient exploration issue in outcome-only end-to-end RL for long-horizon LLM agents, where spurious state–action correlations override genuine reasoning, leading to redundant reasoning steps and illogical action loops.

2. We introduce a novel framework, RLVMR, that provides dense, verifiable rewards for meta-reasoning behaviors like planning, exploration, and reflection, enabling more robust and efficient problem-solving.

3. We achieve SOTA performance on ALFWorld and ScienceWorld, with in-depth analysis confirming reductions in redundant actions and improved generalization to unseen tasks.

## 2 INEFFICIENT EXPLORATION IN LONG-HORIZON AGENTS

This section investigates the phenomenon of "inefficient exploration" in agents designed for long-horizon tasks. We analyze its detrimental effects on performance, which manifest as **brittle efficiency** on previously seen tasks and **poor generalization** to unseen ones.

## 2.1 Experimental Setup

**Benchmarks**   To evaluate foundational capabilities and generalization, we conduct experiments on the widely-used and challenging **ALFWorld** benchmark (Shridhar et al., 2020), which comprises embodied household tasks. To systematically measure generalization, we define three evaluation splits based on the original benchmark:

- **L0** (*seen-L0*): seen task variants and seen task categories;
- **L1** (*unseen-L1*): unseen held-out task variants but seen task categories;
- **L2** (*unseen-L2*): unseen held-out task variants and unseen task categories.

L0 and L1 follow the official benchmark splits. For L2, we further partition ALFWorld by task category, holding out entire categories from training for exclusive use in evaluation.

**Training Paradigms**   We experiment with Qwen2.5-1.5B-Instruct and Qwen2.5-7B-Instruct models using the **ReAct** (Yao et al., 2023) framework, which alternates between reasoning and acting steps. We evaluate two dominant training paradigms:

- **SFT** (Yang et al., 2023; Tang et al., 2023; Xi et al., 2024): A widely adopted paradigm that applies supervised fine-tuning on high-quality expert trajectories.
- **GRPO** (Feng et al., 2025a; Wang et al., 2025b): An end-to-end RL method that optimizes the policy by comparing the final rewards of multiple trajectories sampled from the same initial state.

**Evaluation Metrics**   We assess performance using the following metrics:

- **Success Rate (%, ↑):** The percentage of tasks successfully completed by the agent on each evaluation split.
- **Invalid Action Rate (%, ↓):** The proportion of generated actions that are invalid in the current state, reflecting basic comprehension and error frequency.
- **Repetitive Action Rate (%, ↓):** The percentage of steps where the agent executes a **meaningless repeated action**, as defined in prior work (Yuan et al., 2025; Fu et al., 2025; Feng et al., 2025b). This metric quantifies inefficient exploration, indicating that the agent's policy may be overfitting to familiar action sequences rather than being guided by robust reasoning.

## 2.2 The Inefficient Exploration Problem

While aggregate statistics show that methods like GRPO can improve agent success rates, a closer look at individual trajectories reveals a critical flaw: the **inefficient exploration problem**. Even when an agent successfully completes a task, its path to a solution is often littered with redundant or illogical steps. This behavior, illustrated qualitatively in Appendix A, indicates a gap between achieving a correct outcome and demonstrating robust reasoning. Our large-scale empirical results (Figure 2) quantify the pervasiveness of this issue and expose a fundamental trade-off in current training paradigms.

**SFT creates efficient but brittle policies that fail to generalize.**   Supervised Fine-Tuning (SFT) models achieve high success rates and efficiency on tasks they have seen during training. For instance, the 7B SFT model's success rate on in-distribution tasks (L0) jumps from 23.1% (ReAct baseline) to 63.3%, with a low invalid action rate of 6.2%. However, this performance is brittle. On the most challenging out-of-distribution split (L2), the model's success rate plummets to 37.5%, and its repetitive action rate nearly doubles. This reveals that when faced with novel situations, the agent falls into non-productive loops, demonstrating that SFT teaches mimicry without instilling a generalizable reasoning process.

**GRPO improves generalization but fosters inefficient, flawed reasoning.**   In contrast, reinforcement learning with outcome-only rewards (GRPO) achieves substantially better generalization, with the 7B model attaining success rates of 77.3% on L1 and 52.3% on L2. This success, however, comes at the cost of severe inefficiency, validating our core hypothesis. The agent's performance

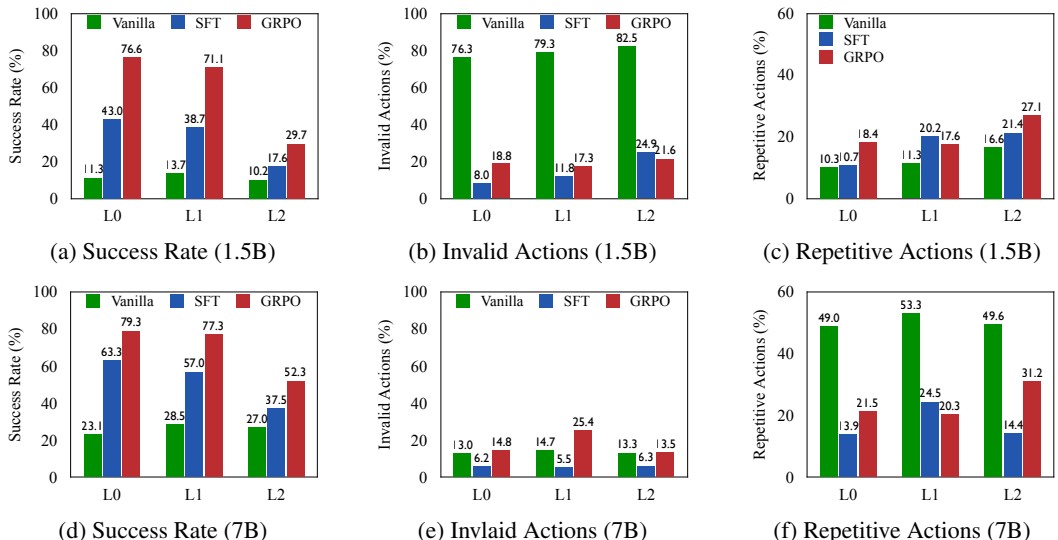

Figure 2: Performance on ALFWorld. While SFT excels on seen tasks (L0) but fails to generalize, GRPO achieves better generalization at the cost of significant inefficiency. This highlights a fundamental trade-off between brittle efficiency and inefficient generalization.

is undermined by high invalid and repetitive action rates across all difficulty levels; on the hardest L2 tasks, the 7B model's repetitive action rate is a staggering 31.2%. By optimizing solely for task success, GRPO reinforces any path to a positive outcome, even those built on illogical steps and inefficient exploration.

**Scaling model size does not fix the underlying reasoning deficiencies.** While scaling from a 1.5B to a 7B model improves overall success rates, it does not resolve this fundamental issue. Notably, while the 7B GRPO model is more successful on L2 tasks than its 1.5B counterpart (52.3% vs. 29.7%), it also exhibits a *higher* repetitive action rate (31.2% vs. 27.1%). This suggests a larger model's enhanced capacity can be misdirected to more effectively exploit flawed strategies rather than to reason more coherently. This finding underscores that the limitation is rooted in the training objective itself, not merely model capacity, and that simply increasing model size is not a panacea.

**Current paradigms force a trade-off between brittle efficiency and inefficient generalization.** Our analysis reveals a core dilemma: SFT produces efficient but brittle policies that fail to generalize, while GRPO achieves generalization at the cost of reinforcing inefficient and logically flawed reasoning. Neither paradigm effectively teaches the agent *how* to reason well. This establishes a clear need for a new framework that moves beyond sparse, outcome-only signals to provide direct, **process-level supervision**. By rewarding coherent and efficient reasoning steps, we can guide agents to not only find solutions but to do so robustly and intelligently — the precise goal of our work.

## 3 METHODOLOGY: RLVMR

Our methodology equips LLM agents with an explicit meta-reasoning framework to mitigate inefficient exploration in complex tasks. As shown in Figure 3, the agent is trained in two phases: an initial SFT stage to bootstrap the agent's meta-reasoning capabilities, followed by a reinforcement learning phase that uses a custom policy optimization algorithm to refine these skills based on task outcomes and process-centric rewards.

**Cold Start: Initial Meta-Reasoning Acquisition via SFT**   To equip the base LLM with the foundational ability to generate structured meta-reasoning, we begin with a supervised fine-tuning phase. This step is crucial, as reasoning patterns learned during subsequent reinforcement learning are heavily influenced by the base model's capabilities. The SFT data is constructed as follows:

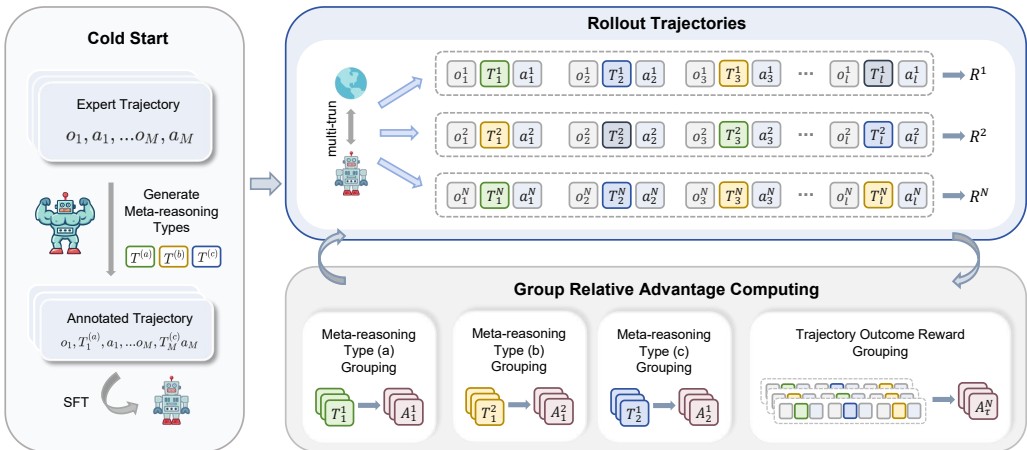

Figure 3: A schematic diagram of the RLVMR framework, which consists of two training phases: cold start and reinforcement learning. Our method provides rule-verifiable feedback signals based on the final outcome and the relative advantages of different types of meta-reasoning behaviors.

1. We collect a dataset of successful task trajectories containing only observation-action pairs.

2. We employ a more powerful teacher model (e.g., GPT-4) to annotate these trajectories with our meta-reasoning tags, inferring the most likely cognitive step preceding each action. This process creates synthetic, reasoning-rich expert demonstrations.

3. The target LLM is fine-tuned on these annotated trajectories, learning to imitate the expert's meta-reasoning and action generation patterns.

## 3.1 META-REASONING FRAMEWORK

We begin by formalizing the agent-environment interaction as a Markov Decision Process. We then introduce a novel meta-reasoning framework that extends existing agent architectures by operationalizing principles from cognitive science.

**Task Formulation as a Markov Decision Process**    We formalize the interaction between an agent and its environment in long-horizon tasks as a Markov Decision Process (MDP). An MDP is defined by a tuple $(S, A, O, F, R)$, where $S$ is the set of environment states, $A$ is the action space, $O$ is the observation space, $F : S \times A \to S$ is the state transition function, and $R : S \times A \to \mathbb{R}$ is the reward function. In our setting, which is tailored for LLM agents, the state, action, and observation spaces $(S, A, O)$ are all represented as natural language sequences over a finite token vocabulary.

At each timestep $t$, the agent's policy $\pi_\theta$ generates a thought process $th_t$ and an action $a_t$ based on the current state $s_t$: $(th_t, a_t) \sim \pi_\theta(\cdot \mid s_t)$. The agent's interaction with the environment produces a trajectory $\tau = \{(o_1, th_1, a_1), (o_2, th_2, a_2), \ldots, (o_n, th_n, a_n)\}$. In many long-horizon tasks, reward signals are sparse, typically provided only as a final outcome reward $R(\tau)$ at the end of an episode. This sparsity poses significant challenges for credit assignment. The agent's objective is to learn an optimal policy $\pi_\theta$ that maximizes the expected cumulative reward:

$$\max_\theta \ \mathbb{E}_{\tau \sim \pi_\theta} \left[ R(\tau) \right]. \tag{1}$$

**Operationalizing Meta-Reasoning in LLM Agents**    Our approach is grounded in metacognitive theory (Martinez, 2006; Lai, 2011), which emphasizes "thinking about thinking". Metacognition comprises two key components: *metacognitive knowledge* (an agent's self-awareness of its own reasoning strategies) and *metacognitive regulation* (the active control of these processes, including planning, monitoring, and adaptive revision). This theoretical lens suggests that for LLM agents to solve complex tasks, they require not just domain knowledge but also the capacity for dynamic planning, self-monitoring, and creative exploration.

To operationalize these principles, we extend the ReAct framework. While ReAct interleaves reasoning and actions (e.g., "Think: ..., Act: ..."), it treats reasoning as a monolithic process. We refine this by introducing a structured set of meta-reasoning tags to explicitly represent distinct cognitive functions. This decouples reasoning from actions and enables fine-grained analysis and supervision. Specifically, we define four meta-reasoning tags, each enclosed in XML-style tags (e.g., `<planning>`), while all actions are contained within the `<action>` tag.

- **Planning (`<planning>`):** Decomposes the task into high-level steps to formulate an overall strategy. Used at the start of a task or when replanning is needed.
- **Exploration (`<explore>`):** Generates hypotheses or options to navigate uncertainty or bottlenecks, encouraging creative problem-solving.
- **Reflection (`<reflection>`):** Reviews history to analyze errors and formulate corrective actions. Typically triggered after unsuccessful attempts.
- **Monitoring (`<monitor>`):** Tracks task progress against the overall plan, ensuring actions remain aligned with subgoals. Applied during routine execution.

## 3.2 META-REASONING-AWARE REWARD SHAPING

During reinforcement learning, we guide the agent with a composite reward signal that combines task completion with the quality of the reasoning process. This signal comprises a sparse outcome reward and a dense, process-based meta-reasoning reward.

**Outcome Reward ($R(\tau)$):** A binary signal awarded at the end of a trajectory: $R(\tau) = r_s$ for task success and $0$ otherwise, where $r_s$ is a positive constant.

**Meta-Reasoning Reward ($r_t^{\mathrm{MR}}$):** A dense reward assigned at each step $t$ to incentivize locally beneficial behaviors.

- **Planning Reward ($r_{\mathrm{planning}}$):** Awarded for a `<planning>` step if the trajectory succeeds.
- **Exploration Reward ($r_{\mathrm{explore}}$):** Awarded if the current action targets a new object or location, discouraging redundancy.
- **Reflection Reward ($r_{\mathrm{reflection}}$):** Awarded if a `<reflection>` step is followed by a corrective action after a sequence of failures.

**Format Reward ($r_t^{\mathrm{format}}$):** A penalty, $-\lambda_{\mathrm{format}}$, is applied if the model's output at step $t$ does not conform to the expected `<tag>...</tag><action>...</action>` structure.

The total step-level reward is the sum of the process-based rewards: $r_t = r_t^{\mathrm{MR}} + r_t^{\mathrm{format}}$.

## 3.3 GROUP RELATIVE POLICY OPTIMIZATION WITH META-REASONING (GRPO-MR)

To effectively leverage our composite reward signal, we introduce Meta-Reasoning Group Policy Optimization (GRPO-MR). GRPO-MR computes a step-level advantage by combining global trajectory performance with local, context-aware reasoning quality.

**Trajectory-level Relative Advantage:** For a batch of $K$ trajectories collected from the same environment, we first calculate a normalized trajectory-level advantage to capture overall performance:

$$A_k^{\mathrm{traj}} = \frac{R(\tau_k) - \mu_R}{\sigma_R}, \tag{2}$$

where $\mu_R$ and $\sigma_R$ are the mean and standard deviation of outcome rewards across the batch.

**Meta-reasoning Level Relative Advantage:** The core of GRPO-MR is the computation of a context-aware advantage. We group all steps within a batch that share the same meta-reasoning tag (e.g., all `<explore>` steps) and normalize their rewards *within* that group:

$$A_{t,\mathrm{tag}}^{\mathrm{MR}} = \frac{r_{t,\mathrm{tag}}^{\mathrm{MR}} - \mu_{\mathrm{tag}}}{\sigma_{\mathrm{tag}}}, \tag{3}$$

Table 1: Performance comparison on the benchmarks. We report the success rate (%) on seen (L0: seen task variants and categories) and unseen (L1: unseen task variants but seen task categories; L2: unseen task variants and categories) task variations. We also report the average cumulative reward (score) on the ScienceWorld benchmark.

| Model | Method | ALFWorld | | | ScienceWorld | | | | | |
| | | L0 | L1 | L2 | L0 | | L1 | | L2 | |
| | | succ. | succ. | succ. | succ. | score | succ. | score | succ. | score |
|---|---|---|---|---|---|---|---|---|---|---|
| GPT-4o | ReAct | 57.3 | 66.0 | 68.8 | 45.4 | 54.3 | 49.2 | 57.0 | 41.0 | 52.0 |
| DeepSeek-V3 | ReAct | 60.2 | 65.9 | 53.9 | 27.3 | 39.1 | 35.2 | 43.0 | 26.5 | 37.1 |
| DeepSeek-R1 | ReAct | 68.8 | 70.2 | 67.3 | 22.2 | 32.0 | 31.4 | 39.5 | 29.1 | 37.9 |
| AgentGym | SFT+RL | 76.6 | 63.3 | - | 46.9 | 56.3 | 33.6 | 45.2 | - | - |
| | ReAct | 11.3 | 13.7 | 10.2 | 1.2 | 9.0 | 0.8 | 7.8 | 0.8 | 7.4 |
| | + SFT | 43.0 | 38.7 | 17.6 | 20.3 | 30.9 | 18.0 | 27.8 | 12.5 | 20.9 |
| | + ETO | 64.1 | 66.4 | 25.8 | 39.1 | 47.3 | 22.7 | 29.8 | 15.6 | 23.4 |
| Qwen2.5-1.5B | + GLIDER | 66.0 | 68.8 | 35.2 | 40.2 | 50.2 | 25.8 | 32.0 | 19.5 | 25.1 |
| | + GRPO | 76.6 | 71.1 | 29.7 | 21.1 | 31.7 | 13.7 | 22.5 | 10.9 | 21.2 |
| | + GiGPO | 86.7 | 83.2 | 48.0 | 25.8 | 35.6 | 15.2 | 22.8 | 4.7 | 11.2 |
| | + RLVMR | **89.1** | **87.9** | **56.3** | **46.9** | **60.3** | **34.4** | **45.2** | **26.5** | **33.9** |
| | ReAct | 23.1 | 28.5 | 27.0 | 7.8 | 17.4 | 11.3 | 19.6 | 6.3 | 16.5 |
| | + SFT | 63.3 | 57.0 | 37.5 | 36.7 | 43.5 | 32.0 | 41.6 | 23.4 | 32.2 |
| | + ETO | 70.3 | 74.2 | 51.6 | 62.5 | 71.2 | 40.6 | 50.4 | 28.1 | 35.0 |
| Qwen2.5-7B | + GLIDER | 75.4 | 74.6 | 53.1 | 62.9 | 68.8 | 41.4 | 52.8 | 25.8 | 32.5 |
| | + GRPO | 79.3 | 77.3 | 52.3 | 49.1 | 61.8 | 30.1 | 43.1 | 26.6 | 34.3 |
| | + GiGPO | 89.5 | 90.2 | 67.2 | 53.4 | 69.2 | 35.2 | 50.7 | 25.8 | 33.2 |
| | + RLVMR | **91.4** | **91.8** | **83.6** | **67.2** | **77.8** | **43.0** | **59.4** | **32.2** | **49.1** |
| | ReAct | 19.5 | 22.3 | 17.6 | 8.6 | 18.8 | 11.7 | 19.9 | 11.7 | 20.3 |
| | + SFT | 62.5 | 60.9 | 39.1 | 39.8 | 47.6 | 30.1 | 39.8 | 22.3 | 32.6 |
| | + ETO | 69.5 | 67.5 | 47.3 | 57.0 | 64.3 | 36.8 | 45.2 | 29.3 | 35.4 |
| Llama3.1-8B | + GLIDER | 72.7 | 73.4 | 50.8 | 64.4 | 71.2 | 38.7 | 53.8 | 28.5 | 35.6 |
| | + GRPO | 73.0 | 70.7 | 45.3 | 45.6 | 55.2 | 28.8 | 40.1 | 25.8 | 33.7 |
| | + GiGPO | 86.0 | 87.1 | 68.8 | 60.2 | 73.5 | 39.1 | 55.2 | 30.1 | 42.3 |
| | + RLVMR | **92.2** | **91.0** | **83.2** | **71.1** | **80.3** | **49.2** | **63.7** | **38.7** | **51.2** |

where $\mu_{\text{tag}}$ and $\sigma_{\text{tag}}$ are the mean and standard deviation of meta-reasoning rewards for all steps with that specific tag. The final step-level advantage $A_t$ is a weighted combination of these two signals:

$$A_t = \alpha \cdot A_k^{\text{traj}} + (1 - \alpha) \cdot A_{t,\text{tag}}^{\text{MR}}, \tag{4}$$

where $\alpha \in [0, 1]$ is a hyperparameter balancing the influence of the global outcome and local reasoning quality. Finally, we optimize the policy $\pi_\theta$ using a clipped surrogate objective with KL divergence regularization:

$$\mathcal{L}_{\text{final}} = \mathbb{E}_t \left[ \min \left( r_t(\theta) A_t, \text{clip}(r_t(\theta), 1 - \epsilon, 1 + \epsilon) A_t \right) \right] - \lambda_{\text{KL}} D_{\text{KL}}(\pi_\theta \| \pi_{\text{ref}}), \tag{5}$$

where $r_t(\theta)$ is the importance sampling ratio, $\epsilon$ is the clipping hyperparameter, and $\lambda_{\text{KL}}$ controls the KL penalty against a reference policy $\pi_{\text{ref}}$.

# 4 EXPERIMENT

## 4.1 MAIN RESULTS

In this section, we present the core experimental results to evaluate the effectiveness of our proposed RLVMR. In addition to **ALFWorld**, we also conduct experiments on **ScienceWorld** (Wang et al., 2022), which focuses on text-based scientific experimentation.

We compare our approach with two major categories of advanced RL training methods in addition to SFT: (1) Offline RL, including (i) **ETO** (Song et al., 2024), which iteratively refines actions using step-level feedback along trajectories; (ii) **GLIDER** (Hu et al., 2025b), which decomposes complex

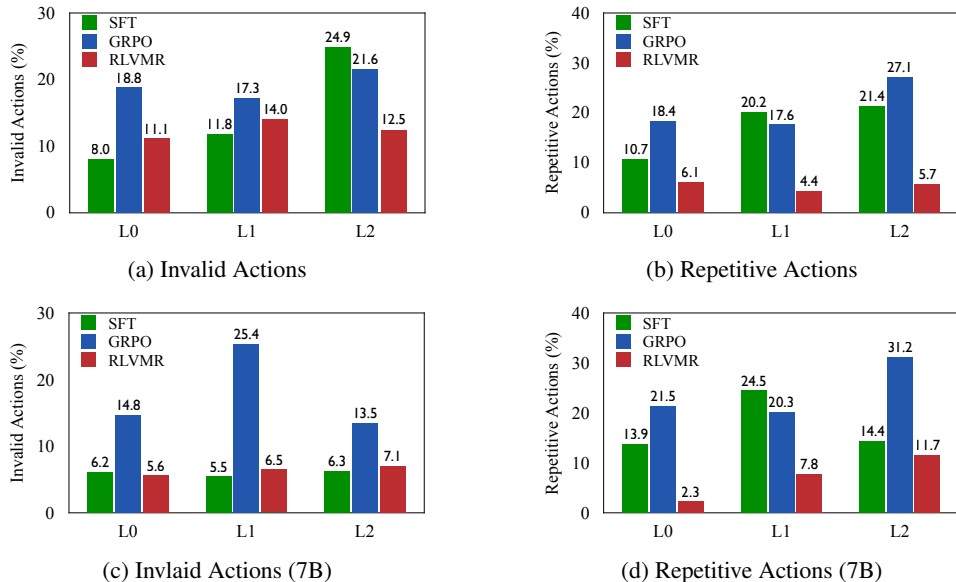

Figure 4: Exploration efficiency of RLVMR compared to SFT and GRPO baselines on ALFWorld.

tasks into coherent sub-tasks to improve transferability. (2) Online End-to-end RL, including (iii) **Multi-turn GRPO** (Wang et al., 2025b), which adapts the original GRPO (Shao et al., 2024) for online multi-turn RL tasks; (iv) **GiGPO** (Feng et al., 2025b), which introduces a two-level structure for finer-grained credit assignment. For broader comparison, we also report the performance of GPT-4o, DeepSeek-V3/R1, and AgentGym (Xi et al., 2024). Detailed information is provided in Appendix B.

**RLVMR achieves new SOTA performance across all benchmarks and model sizes.** As listed in Table 1, our RLVMR framework consistently sets a new standard for performance, outperforming all baseline methods on both ALFWorld and ScienceWorld. With the Qwen-7B model, RLVMR achieves success rates of 91.4% on seen ALFWorld tasks and 67.2% on seen ScienceWorld tasks, surpassing the next-best method, GiGPO. This consistent superiority highlights the broad applicability and effectiveness of integrating verifiable meta-reasoning rewards into the RL training loop, leading to more capable and successful agents.

**Rewarding meta-reasoning significantly enhances generalization to unseen tasks.** A primary contribution of this work is addressing the inefficient exploration issue to improve generalization. Our results validate this claim, showing that RLVMR excels in novel scenarios, especially on the most challenging Unseen-L2 split, which involves entirely new task categories. On ALFWorld's L2 split, our 7B model reaches an impressive 83.6% success rate, a substantial 16.4 percentage point improvement over the strongest baseline (GiGPO). Similarly, on ScienceWorld's L2 split, RLVMR outperforms all other methods. This demonstrates that by learning **how** to reason effectively—rather than just memorizing solutions—our agent develops more robust and transferable problem-solving skills, leading to superior performance on unfamiliar challenges.

## 4.2 ANALYSIS

Our analysis reveals that RLVMR's verifiable meta-reasoning rewards lead to superior exploration and training efficiency, enabling the agent to find more direct solutions with greater stability than strong baselines. Unless otherwise stated, we report results based on Qwen2.5-1.5B on ALFWorld.

**Exploration Efficiency** We analyze agent exploration efficiency by measuring invalid and repetitive actions (Figure 4). RLVMR's verifiable meta-reasoning rewards cultivate more efficient problem-solving strategies, significantly reducing flawed or redundant steps. On seen tasks, our 1.5B model slashes the invalid action rate from 18.1% (GRPO) to 11.1% and the repetitive action rate from 18.4%

to 6.1%. This efficiency gain is robustly maintained on novel challenges; while GRPO's repetitive action rate worsens on the hardest unseen tasks (from 21.4% to 27.1%), RLVMR's rate remains controlled at 5.7%. This demonstrates that RLVMR learns generalizable problem-solving principles rather than overfitting to familiar paths.

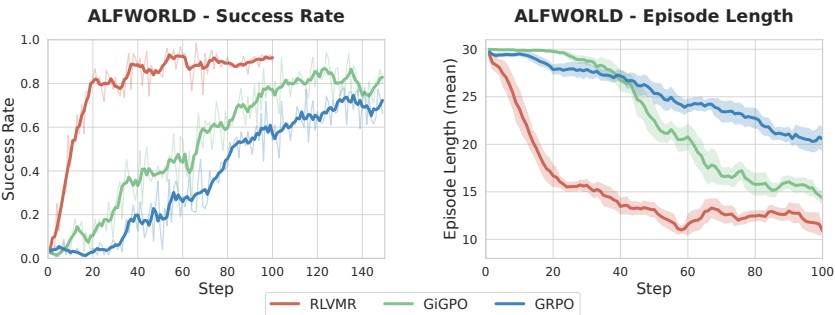

Figure 5: Success rate and step count curves of different approaches on ALFWorld during RL training.

**Training Efficiency**  We evaluate training efficiency via learning stability (convergence) and policy quality (action sequence length) in Figure 5. Agents trained with RLVMR learn more direct solutions and converge faster and more stably than baselines. In contrast, baselines like GRPO are unstable and produce longer solution paths. This stems from its process-level rewards, which provide a clearer and more robust learning signal that prevents inefficient and unproductive loops.

## 5 RELATED WORK

**LLM Reinforcement Learning**  RL is widely used to align LLMs with human preferences (RLHF, DPO) (Ouyang et al., 2022; Rafailov et al., 2023). Beyond alignment, RL has been applied to improve reasoning and emotional intelligence (Hu et al., 2025a; Muennighoff et al., 2025; Wang et al., 2025a). Group-based methods such as GRPO, Dr.GRPO, and DAPO estimate advantages from multiple samples of the same prompt, removing the critic and improving efficiency over actor-critic approaches like PPO (Feng et al., 2025a; Liu et al., 2025; Yu et al., 2025; Schulman et al., 2017). These methods achieve strong results on mathematical reasoning, search, and tool use (Yu et al., 2025; Hu et al., 2025a). However, applying RL to multi-turn, long-horizon tasks remains difficult due to sparse, delayed rewards – a challenge we address (Wang et al., 2025b).

**LLM Agents**  LLMs increasingly act as agents for code generation, web interaction, embodied control, and affective tasks (Huang et al., 2023; Zhang et al., 2024; Bai et al., 2024; Agashe et al., 2024; Abuelsaad et al., 2024; Zeng et al., 2024; Qiao et al., 2024; Fu et al., 2025; Zhang et al., 2025). Early systems relied on prompting and external tools (e.g., ReAct) (Yao et al., 2023; Shinn et al., 2023), but smaller models often lack strong reasoning; SFT can improve decisions (Zhang & Zhang, 2024; Xi et al., 2024; Qin et al., 2024). Other work studies single-step or offline RL (Yu et al., 2024; Xiong et al., 2024; Zhou & Zanette, 2024), while recent efforts train agents end to end with online RL, learning directly from interaction and reducing reliance on complex data preparation or step-level reward models (Wang et al., 2025b; Feng et al., 2025b). Despite progress, fine-grained credit assignment and generalization remain challenging (Wang et al., 2025b). We employ reward shaping grounded in verifiable meta-cognitive behaviors to promote effective reasoning and robustness.

## 6 CONCLUSION

We tackled the challenge of inefficient exploration in long-horizon agents by introducing RLVMR, a new framework that guides agents using process-level supervision. Instead of relying solely on sparse success-based rewards, RLVMR provides dense, verifiable feedback for key reasoning behaviors like planning, exploration, and reflection. Our approach combines a lightweight initialization phase with end-to-end training to develop more effective and adaptable agents. Experiments on ALFWorld and ScienceWorld show that RLVMR achieves state-of-the-art performance, with better generalization to

new tasks and noticeable improvements in reasoning quality—fewer redundant actions and better recovery from mistakes. These results highlight the value of directly supervising reasoning steps. Future research could extend RLVMR to multi-modal environments, explore adaptive reward mechanisms that dynamically adjust to task complexity, and apply the framework to real-world domains such as robotics and software engineering.

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

## A    CASE STUDY OF INEFFICIENT EXPLORATION

The following is a trajectory segment of a vanilla GRPO-trained agent performing a novel task——
*put two keychains in the safe* — in ALFWorld (corresponding to our L2 split). By step 7, the agent
has already arrived at dresser 1. However, we observe that in the subsequent steps, the agent falls into
a sequence of inefficient decisions: its next intention is to find the second keychain, yet it persistently
tries to go to dresser 1 for several steps, disregarding the fact that it is already there. This indicates
that its policy mainly reflects the action distribution present in the training data, rather than allowing
the reasoning process itself to truly regulate decision-making. Although the agent can form relatively
effective action strategies for completing tasks, its capacity for critically evaluating its own behavior
and understanding the underlying task requirements remains limited. This suggests that the agent has
not truly acquired the reasoning patterns necessary for robust task-solving.

---

**A trajectory segment of an agent trained with GRPO**

**Step 7**
**Observation:** You arrive at dresser 1. On the dresser 1, you see a box 1, a creditcard 1, a
keychain 2, a keychain 1, a laptop 1, a statue 1, and a tissuebox 2.
**Reasoning:** The current situation is that I am at dresser 1, which has multiple items including
a keychain 1, a keychain 2, a creditcard 1, etc. To complete the task of putting two keychains
in the safe, I need to select one of these keychains from the dresser.
**Action:** take keychain 1 from dresser 1

**Step 8**
**Observation:** You pick up the keychain 1 from the dresser 1.
**Reasoning:** The current situation is that I am at dresser 1 with the keychain 1 in my
possession. To put two keychains in the safe, I need to find another keychain and then move
both keychains to the safe.
**Action:** go to dresser 1

**Step 9**
**Observation:** Nothing happens.
**Reasoning:** The player has been in the room for several steps but has not yet interacted with
the safe or taken any keychains. The safe is closed initially, which might need interaction to
open it.
**Action:** go to dresser 1

**Step 10**
**Observation:** Nothing happens.
**Reasoning:** The previous steps were not successful in opening the safe or taking the
keychains. Let's try another approach.
**Action:** go to dresser 1

**Step 11**
**Observation:** Nothing happens.
**Reasoning:** The user has taken all the steps necessary to obtain both keychains but has yet to
actually put them into the safe. The last known location of the keychains was on the dresser
before the attempt to open the safe failed (in Observation 8). To complete the task, the user
needs to move one or both keychains to the safe before attempting to open it again.
**Action:** move keychain 1 to safe 1

**Observation:** Nothing happens.

---

This qualitative example highlights how an ostensibly "successful" policy can still waste a large
number of steps on redundant or illogical moves.

# B    SETUP DETAILS

## B.1    DATASET DETAILS

**ALFWorld** is a household task environment that requires agents to explore rooms and employ common-sense reasoning to accomplish tasks, such as "put the pencil on the desk". The environment provides feedback on whether the agent successfully completes the task within a given number of steps. ALFWorld describes the environment in purely textual form and supplies a reward signal indicating only the final task completion status.

**ScienceWorld** is a text-based virtual environment designed as a comprehensive testbed for evaluating and enhancing scientific reasoning abilities in AI systems. It features tasks spanning 10 scientific domains and 30 subcategories, simulating a broad range of experiments found in elementary science curricula, including state changes of matter, measurement, electricity, life sciences, plant growth, chemical reactions, genetics, among others. Each task involves multiple subgoals, and the final reward is computed based on the completion of these subgoals. However, to better reflect real-world scenarios, we only use the final reward and disregard intermediate rewards. Notably, some tasks in ScienceWorld require agents to make conclusive judgments based on experimental outcomes or common sense; a task is considered successful only if the agent provides the correct final answer.

Both ALFWorld and ScienceWorld offer "seen" and "unseen" variants for evaluating generalization capabilities. To further assess the agents' robustness and generalization, we define three difficulty levels (L0, L1, L2), with L2 comprising entirely held-out task types. Specifically, for ALFWorld, we designate *Cool & Place* and *Pick Two & Place* as held-out tasks; for ScienceWorld, the final task type of each topic is reserved for unseen evaluation.

In the ALFWorld environment, since only the final task success signal is provided, we evaluate model performance using the average success rate (**succ.**). In contrast, the ScienceWorld environment offers more fine-grained step rewards, enabling the agent to obtain immediate rewards based on the importance of the steps completed, even without achieving the final goal. Therefore, in addition to the average success rate (**succ.**), we also report the average cumulative reward (**score**).

## B.2    IMPLEMENTATION DETAILS

We conducted experiments on both the Qwen2.5-1.5B-Instruct and Qwen2.5-7B-Instruct models. During the cold start phase, we set the batch size per GPU to 16, used a learning rate of $1 \times 10^{-5}$, and trained for 5 epochs. For the RL phase, we adopted the veRL framework with necessary modifications. The batch size per GPU was also set to 16. At each training step, we sampled from 16 distinct environments, with each environment rolling out 8 trajectories.

The weights for outcome advantage and meta-reasoning advantage were both set to $0.5$ by default. To penalize outputs that did not adhere to the required format, we applied a reward penalty of $-0.1$, where an output was considered valid only if it included at least one meta-reasoning tag (e.g., $\langle$reflection$\rangle$) and one action tag (e.g., $\langle$action$\rangle$). The KL regularization coefficient was set to $0.01$. For all environments, the maximum number of steps per episode was fixed at 30. In the cold-start phase, we performed supervised fine-tuning on 200 trajectories for 5 epochs. In the RL training stage, our method was run for 100 epochs, whereas RL-based baselines were trained for 150 epochs.

AgentGym is trained on Llama-2-Chat-7B, first with behavior cloning on the AgentTraj (Xi et al., 2024) dataset from multiple environments, and then further improved via exploration and self-evolution on a broader instruction set.

# C    DETAILED EXPERIMENT RESULTS

We further report the success rates of different methods on various tasks in ALFWorld. Table 2 provides the results using the Qwen2.5-1.5B model as the base model, while Table 3 presents the results using the Qwen2.5-7B model. Additionally, we also evaluated another model from a different family, Llama3.1-8B, and the results are shown in Table 4. As shown in the tables, RLVMR generally outperforms other methods across all tasks, and particularly exhibits strong performance in more

complex tasks. This demonstrates that RLVMR, by rewarding high-quality reasoning behaviors, significantly enhances the robustness and adaptability of agents in multi-step interactions.

| Model | Method | Pick | Look | Clean | Heat | Cool | Pick2 | All |
|---|---|---|---|---|---|---|---|---|
| Qwen2.5-1.5B | ReAct | 23.1 | 18.3 | 10.8 | 8.7 | 3.5 | 0.0 | 13.7 |
| | +SFT | 43.2 | 42.0 | 35.9 | 33.2 | 29.4 | 29.7 | 38.7 |
| | +ETO | 73.6 | 46.3 | 66.2 | 68.3 | 62.8 | 55.6 | 66.4 |
| | +GLIDER | 78.8 | 58.2 | 63.6 | 73.7 | 61.6 | 66.1 | 68.8 |
| | +GRPO | 80.3 | 55.6 | 88.1 | 76.2 | 62.0 | 72.1 | 71.1 |
| | +GiGPO | 92.8 | 66.5 | 90.7 | 90.9 | 80.2 | 73.8 | 83.2 |
| | +RLVMR | 95.2 | 78.8 | 91.2 | 90.2 | 83.9 | 77.6 | 87.9 |

Table 2: Success rates on ALFWorld using Qwen2.5-1.5B model.

| Model | Method | Pick | Look | Clean | Heat | Cool | Pick2 | All |
|---|---|---|---|---|---|---|---|---|
| Qwen2.5-7B | ReAct | 43.1 | 33.2 | 18.7 | 16.4 | 20.2 | 12.8 | 28.5 |
| | +SFT | 70.8 | 63.0 | 61.1 | 46.3 | 49.7 | 33.2 | 57.0 |
| | +ETO | 88.2 | 70.5 | 82.3 | 83.6 | 71.0 | 51.2 | 74.2 |
| | +GLIDER | 89.6 | 72.1 | 83.9 | 81.6 | 69.5 | 53.0 | 74.6 |
| | +GRPO | 90.2 | 76.7 | 86.0 | 80.1 | 68.3 | 56.4 | 77.3 |
| | +GiGPO | 91.7 | 85.9 | 93.3 | 90.3 | 89.0 | 83.6 | 90.2 |
| | +RLVMR | 95.3 | 88.2 | 90.1 | 92.4 | 89.8 | 86.7 | 91.8 |

Table 3: Success rates on ALFWorld using Qwen2.5-7B model.

| Model | Method | Pick | Look | Clean | Heat | Cool | Pick2 | All |
|---|---|---|---|---|---|---|---|---|
| Llama3.1-8B | ReAct | 40.3 | 30.1 | 17.8 | 13.9 | 19.5 | 9.3 | 22.3 |
| | +SFT | 70.8 | 69.0 | 58.6 | 47.7 | 58.9 | 40.4 | 60.9 |
| | +ETO | 83.3 | 64.5 | 76.9 | 73.0 | 66.4 | 46.2 | 67.5 |
| | +GLIDER | 87.7 | 71.2 | 78.0 | 79.5 | 68.2 | 49.7 | 73.4 |
| | +GRPO | 87.0 | 75.9 | 82.8 | 74.0 | 67.2 | 55.0 | 70.7 |
| | +GiGPO | 90.3 | 87.5 | 90.1 | 85.2 | 83.6 | 82.5 | 87.1 |
| | +RLVMR | 93.5 | 90.0 | 86.5 | 91.5 | 86.5 | 83.5 | 91.0 |

Table 4: Success rates on ALFWorld using Llama3.1-8B model.

# D    PSEUDOCODE OF RLVMR

We present the pseudocode for the RLVMR training procedure in Algorithm 1, and the pseudocode for computing the relative advantage of composite groups in Algorithm 2. Additionally, we provide the pseudocode for computing meta-reasoning rewards in Algorithm 3.

# E    TRAINING CURVES ON SCIENCEWORLD

We also report the success rate curves and average step counts of different RL training methods on ScienceWorld, as shown in Figure 6.

Counterintuitively, when training GRPO or GiGPO on ScienceWorld, the average action steps do not decrease as success rates improve; in some cases, the number even rises. This may be because ScienceWorld tasks require the agent not only to plan, explore, and reflect, but also to connect scientific theories to concrete actions, which smaller models may not perform sufficiently well. Early in training, agents often terminate trajectories early with incorrect answers before sufficient experimentation. As training progresses, these unproductive trajectories are reduced, leading to an

---

**Algorithm 1** RLVMR: Reinforcement Learning with Verifiable Meta-Reasoning Rewards

---

**Require:** Policy $\pi_\theta$, Environment $\mathcal{E}$, Reward function $R$, Hyperparameter $\lambda_{\text{meta}}$
**Ensure:** Optimized policy parameters $\theta$
 1: **for** iteration $t = 1, 2, \ldots, T$ **do**
 2:      Initialize trajectory set $\mathcal{D} = \emptyset$
 3:      **for** episode $i = 1, 2, \ldots, N$ **do**
 4:          Get initial state from environment $s_0^{(i)} \sim \mathcal{E}$
 5:          Initialize trajectory $\tau^{(i)} = \{\}$
 6:          $t \leftarrow 0$
 7:          **while** episode not terminated **do**
 8:              Sample action $a_t^{(i)} \sim \pi_\theta(\cdot|s_t^{(i)})$
 9:              Execute action and observe $s_{t+1}^{(i)}, r_t^{(i)} = \mathcal{E}(s_t^{(i)}, a_t^{(i)})$
10:              // Extract reasoning type tag
11:              $\text{tag}_t^{(i)} \leftarrow \text{ExtractReasoningTag}(a_t^{(i)})$
12:              // $\{\langle\text{planning}\rangle, \langle\text{explore}\rangle, \langle\text{reflection}\rangle, \langle\text{monitor}\rangle\}$
13:              $\tau^{(i)} \leftarrow \tau^{(i)} \cup \{(s_t^{(i)}, a_t^{(i)}, r_t^{(i)}, \text{tag}_t^{(i)})\}$
14:              $t \leftarrow t + 1$
15:          **end while**
16:          // Compute Outcome Reward
17:          $R_{\text{outcome}}^{(i)} \leftarrow R(\tau^{(i)})$
18:      **end for**
19:      // Compute Meta Reasoning Rewards for all trajectories
20:      **for** episode $i = 1, 2, \ldots, N$ **do**
21:          **for** each step $t$ in $\tau^{(i)}$ **do**
22:              $r_{\text{meta},t}^{(i)} \leftarrow \text{ComputeMetaReward}(a_t^{(i)}, \text{tag}_t^{(i)}, \tau^{(i)}, R_{\text{outcome}}^{(i)})$
23:              $\tau^{(i)}[t] \leftarrow \tau^{(i)}[t] \cup \{r_{\text{meta},t}^{(i)}\}$           $\triangleright$ Attach meta-reasoning reward to step $t$
24:          **end for**
25:          $\mathcal{D} \leftarrow \mathcal{D} \cup \{(\tau^{(i)}, R_{\text{outcome}}^{(i)})\}$
26:      **end for**
27:      // Compute group relative advantage
28:      $\{A^{(i)}\}_{i=1}^N \leftarrow \text{ComputeGroupRelativeAdvantage}(\mathcal{D})$
29:      // Group Relative Policy Optimization
30:      **for** update step $k = 1, 2, \ldots, K$ **do**
31:          Compute policy gradient: $\nabla_\theta J(\theta) = \mathbb{E}_{\tau \sim \mathcal{D}}[\nabla_\theta \log \pi_\theta(a|s) \cdot A]$
32:          Update policy parameters: $\theta \leftarrow \theta + \alpha \nabla_\theta J(\theta)$
33:      **end for**
34: **end for**

---

increase in average action steps as agents perform more comprehensive experiments. This reveals a limitation of RL: while it can align LLM behavior with the environment, its effectiveness is constrained by the foundation model's capabilities. Our method mitigates this by applying a cold start phase, allowing the foundation model to acquire essential environmental knowledge. As a result, action steps on ScienceWorld are more stable and exhibit reliable convergence.

## F  PROMPTS

Below are the prompts we used in the ALFWorld and ScienceWorld environments.

---

**Prompt Template for ALFWorld Enviroment**

You are an expert agent operating in the **ALFRED Embodied Environment**. Your task is to:
{task_description}
**Prior to this step, you have already taken {step_count} step(s).**

---

---

**Algorithm 2** Step-Level Group Relative Advantage Computation

---

**Require:** Trajectory data $\mathcal{D} = \{(\tau^{(i)}, R_{\text{outcome}}^{(i)})\}_{i=1}^{N}$, Weight $\lambda_{\text{meta}}$
**Ensure:** Advantage estimates $\{A^{(i)}\}_{i=1}^{N}$
1: // ===== Outcome Advantage Computation =====
2: Group by enviroment index: $\mathcal{G}_{\text{outcome}} = \{g_j\}$ where $g_j = \{i : \text{env\_idx}^{(i)} = j\}$
3: **for** each group $g_j \in \mathcal{G}_{\text{outcome}}$ **do**
4:     Compute group mean: $\mu_j = \frac{1}{|g_j|} \sum_{i \in g_j} R_{\text{outcome}}^{(i)}$
5:     Compute group std: $\sigma_j = \sqrt{\frac{1}{|g_j|} \sum_{i \in g_j} (R_{\text{outcome}}^{(i)} - \mu_j)^2}$
6:     **for** $i \in g_j$ **do**
7:         $A_{\text{outcome}}^{(i)} = \frac{R_{\text{outcome}}^{(i)} - \mu_j}{\sigma_j + \epsilon}$
8:     **end for**
9: **end for**
10: // ===== Meta Reasoning Advantage Computation =====
11: Group by (enviroment index, reasoning tag): $\mathcal{G}_{\text{meta}} = \{g_{j,k}\}$
12: where $g_{j,k} = \{i : \text{env\_idx}^{(i)} = j \wedge \text{tag}^{(i)} = k\}$
13: **for** each group $g_{j,k} \in \mathcal{G}_{\text{meta}}$ **do**
14:     Compute group mean: $\mu_{j,k} = \frac{1}{|g_{j,k}|} \sum_{i \in g_{j,k}} r_{\text{meta}}^{(i)}$
15:     Compute group std: $\sigma_{j,k} = \sqrt{\frac{1}{|g_{j,k}|} \sum_{i \in g_{j,k}} (r_{\text{meta}}^{(i)} - \mu_{j,k})^2}$
16:     **for** $i \in g_{j,k}$ **do**
17:         $A_{\text{meta}}^{(i)} = \frac{r_{\text{meta}}^{(i)} - \mu_{j,k}}{\sigma_{j,k} + \epsilon}$
18:     **end for**
19: **end for**
20: // ===== Final Advantage Combination =====
21: **for** $i = 1, 2, \ldots, N$ **do**
22:     $A^{(i)} = A_{\text{outcome}}^{(i)} + \lambda_{\text{meta}} \cdot A_{\text{meta}}^{(i)}$
23: **end for**
        **return** $\{A^{(i)}\}_{i=1}^{N}$

---

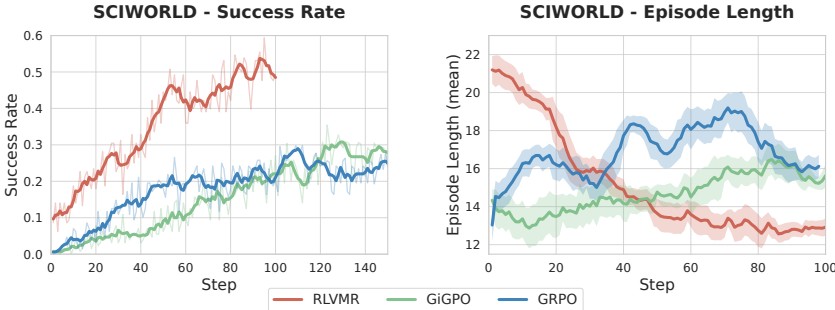

Figure 6: Success rate and step count curves of different approaches on ScienceWorld during RL training.

Below are the most recent {history_length} observations and the corresponding actions you took: {action_history}
You are now at step {current_step} and your current observation is: {current_observation}

**Your admissible actions of the current situation are:** {admissible_actions}.

**Your previous overall plan is:** {planning}. Please strictly adhere to your plan.

---

**Algorithm 3** Meta-Reasoning Reward Computation

---

**Require:** Action $a_t$, State $s_t$, Reasoning tag $\text{tag}_t$, Trajectory $\tau$, Outcome reward $R(\tau)$
**Require:** Reward hyperparameters $\{r_{\text{plan}}, r_{\text{explore}}, r_{\text{reflect}}\}$, discount factor $\gamma$
**Ensure:** Meta-reasoning reward $r_{\text{meta},t}$
 1: $r_{\text{meta},t} \leftarrow 0$
 2: $\text{valid}_t \leftarrow \text{IsActionValid}(a_t)$
 3: **if** $\text{valid}_t = \text{False}$ **then**
 4:     **return** $0$                                    ▷ Invalid actions receive no reward
 5: **end if**
 6: **if** $\text{tag}_t = \langle\text{planning}\rangle$ **then**
 7:     **if** $R(\tau) > 0$ **then**                      ▷ Planning rewarded only on successful trajectories
 8:         $k \leftarrow \text{NumPlanningAfter}(t, \tau)$
 9:         $r_{\text{meta},t} \leftarrow r_{\text{plan}} \cdot \gamma^k$
10:     **else**
11:         $r_{\text{meta},t} \leftarrow 0$
12:     **end if**
13: **end if**
14: **if** $\text{tag}_t = \langle\text{explore}\rangle$ **then**
15:     $\text{isRepeated} \leftarrow \text{False}$
16:     **for** $t' = 0$ to $t-1$ **do**
17:         Extract transition $(s_{t'}, a_{t'}, s_{t'+1})$
18:         **if** $(s_t, a_t, s_{t+1}) = (s_{t'}, a_{t'}, s_{t'+1})$ **then**
19:             $\text{isRepeated} \leftarrow \text{True}$
20:             **break**
21:         **end if**
22:     **end for**
23:     **if** $\text{isRepeated} = \text{False}$ **then**
24:         $r_{\text{meta},t} \leftarrow r_{\text{explore}}$                           ▷ Novel transition
25:     **else**
26:         $r_{\text{meta},t} \leftarrow 0$
27:     **end if**
28: **end if**
29: **if** $\text{tag}_t = \langle\text{reflection}\rangle$ **then**
30:     **if** $t > 0$ **then**
31:         Extract previous transition $(s_{t-1}, a_{t-1})$
32:         $\text{valid}_{t-1} \leftarrow \text{IsActionValid}(a_{t-1})$
33:         **if** $\text{valid}_{t-1} = \text{False}$ **and** $(s_t, a_t) \neq (s_{t-1}, a_{t-1})$ **then**
34:             $r_{\text{meta},t} \leftarrow r_{\text{reflect}}$                       ▷ Effective reflection
35:         **else**
36:             $r_{\text{meta},t} \leftarrow 0$
37:         **end if**
38:     **else**
39:         $r_{\text{meta},t} = 0$
40:     **end if**
41: **end if**
        **return** $r_{\text{meta},t}$

---

Now it's your turn to take an action, following these steps:

1. **First, reason using *ONLY ONE* tag pair and express your reasoning in *one concise, brief sentence***:

   - `<planning>` Plan or replan the entire task by breaking it down into high-level steps. Focus on outlining the full sequence required to complete the overall task, not just the immediate next action. Use this at the beginning of complex tasks or whenever the previous plan is incorrect or insufficient. It is necessary to list all the points separately. eg, step 1: xxx, step 2: xxx, step 3: xxx, etc.

- `<explore>` When results are unexpected or information is lacking, use current observations to think outside the box and list as many possible locations, items, or actions as possible. Use this approach when facing obstacles that require creative and innovative thinking.
- `<reflection>` Analyze the reasons for errors in task execution and correct them by exploring alternative approaches. 'No known action matches that input.' indicates the action is invalid. This is typically used when several consecutive actions yield no substantial progress.
- `<monitor>` Continuously track the current progress and history of reasoning and execution throughout the task. Recall the current subgoal and consider the next concrete action, ensuring agent alignment with the overall plan. Typically used when task outcomes are as expected and no other mode of reasoning is required.

2. **After your reasoning, you *MUST* select and present an admissible action for the current step within `<action>...</action>` tags.**

   Specify the next action the agent should take to progress toward the task goal, following these guidelines:

   (a) **Object and Receptacle References:** Use specific identifiers:
      - `[obj id]` for objects (e.g., apple 1).
      - `[recep id]` for receptacles (e.g., countertop 1).

   (b) **Action Validity:** Follow the exact format below. Any deviation renders the action invalid:
      - Valid actions:    `go to [recep id]`,    `take [obj id] from [recep id]`,    `put [obj id] in/on [recep id]`,    `open/close [recep id]`,    `use [obj id]`,    `heat/cool/clean [obj id] with [recep id]`.

---

**Prompt Template for ScienceWorld Environment**

You are an expert agent operating in the **ScienceWorld** environment, which is a text-based virtual environment centered around accomplishing tasks from the elementary science curriculum.

**Your current task is:** {task_description}

**Prior to this step, you have already taken {step_count} step(s).**
Below are the most recent {history_length} observations and the corresponding actions you took: {action_history}
You are now at step {current_step} and your current observation is: {current_observation}

**Here are the actions you may take:**

- {"action":  "open OBJ", "description":  "open a container"}
- {"action":  "close OBJ", "description":  "close a container"}
- {"action":  "activate OBJ", "description":  "activate a device"}
- {"action":  "deactivate OBJ", "description": "deactivate a device"}
- {"action":  "connect OBJ to OBJ", "description": "connect electrical components"}
- {"action":  "disconnect OBJ", "description": "disconnect electrical components"}

- `{"action": "use OBJ [on OBJ]", "description": "use a device/item"}`
- `{"action": "look around", "description": "describe the current room"}`
- `{"action": "look at OBJ", "description": "describe an object in detail"}`
- `{"action": "look in OBJ", "description": "describe a container's contents"}`
- `{"action": "read OBJ", "description": "read a note or book"}`
- `{"action": "move OBJ to OBJ", "description": "move an object to a container"}`
- `{"action": "pick up OBJ", "description": "move an object to the inventory"}`
- `{"action": "put down OBJ", "description": "drop an inventory item"}`
- `{"action": "pour OBJ into OBJ", "description": "pour a liquid into a container"}`
- `{"action": "dunk OBJ into OBJ", "description": "dunk a container into a liquid"}`
- `{"action": "mix OBJ", "description": "chemically mix a container"}`
- `{"action": "go to LOC", "description": "move to a new location"}`
- `{"action": "eat OBJ", "description": "eat a food"}`
- `{"action": "flush OBJ", "description": "flush a toilet"}`
- `{"action": "focus on OBJ", "description": "signal intent on a task object"}`
- `{"action": "wait", "description": "take no action for 10 iterations"}`
- `{"action": "wait1", "description": "take no action for 1 iteration"}`
- `{"action": "task", "description": "describe current task"}`
- `{"action": "inventory", "description": "list your inventory"}`

**Your previous overall plan is:** {`planning`}.
Please strictly adhere to your plan.

Now it's your turn to take an action, following these steps:

1. **First, reason using *ONLY ONE* tag pair and express your reasoning in *one concise, brief sentence*:**

   - `<planning>`
     Plan or replan the entire task by breaking it down into high-level steps. Focus on outlining the full sequence required to complete the overall task, not just the immediate next action.
     Use this at the beginning of complex tasks or whenever the previous plan is incorrect or insufficient.
     It is necessary to list all the points separately. eg, step 1: xxx, step 2: xxx, step 3: xxx, etc.

- `<explore>`
  When results are unexpected or information is lacking, use current observations to think outside the box and list as many possible locations, items, or actions as possible.
  Use this approach when facing obstacles that require creative and innovative thinking.
- `<reflection>`
  Analyze the reasons for errors in task execution and correct them by exploring alternative approaches. 'No known action matches that input.' indicates the action is invalid.
  This is typically used when several consecutive actions yield no substantial progress.
- `<monitor>`
  Continuously track the current progress and history of reasoning and execution throughout the task. Recall the current subgoal and consider the next concrete action, ensuring agent alignment with the overall plan.
  Typically used when task outcomes are as expected and no other mode of reasoning is required.

2. **After your reasoning, you *MUST* select and present an appropriate action for the current step within `<action>...</action>` tags.**

## G  CASE STUDY OF RLVMR

**Case Study**   Below is a trajectory segment of an agent trained with RLVMR on ALFWorld, performing a novel task: *find two soapbars and put them in the cabinet*. We observe that, guided by meta-cognitive rewards, the agent demonstrates a more comprehensive and meticulous reasoning process, exhibiting significantly more adaptive reasoning patterns. At step 13, the agent is already carrying one soapbar and is actively searching for the second. Lacking explicit information about the soapbars' location, the agent engages in structured and efficient exploration by systematically hypothesizing possible locations via exploratory actions. Concurrently, it keeps track of its current sub-goal, thereby avoiding the task confusion commonly seen in SFT-trained agents.

Notably, in step 14, the agent experiences a brief hallucination, leading to an ineffective action. However, at step 15, rather than falling into an error loop or blindly continuing, the agent immediately triggers reflective behavior, proposing to "double-check my inventory" to collect more information and guard against the possibility of having already acquired both soapbars. This demonstrates a more rigorous and robust reasoning and decision-making process. Experimental results show that such meta-cognitively driven behavioral patterns enhance agent performance and generalization on previously unseen tasks compared to baselines.

---

**A trajectory segment of an agent trained with RLVMR**

**Step 13**
**Observation:** You arrive at sinkbasin 2. On the sinkbasin 2, you see nothing.
**Reasoning:** <explore>Since there are no soapbars at sinkbasin 2, I should broaden my search to other potential locations. The next logical area to check is the toilet, as soapbars could potentially be found there.</explore>
**Action:** go to toilet 1

**Step 14**
**Observation:** You arrive at toilet 1. On the toilet 1, you see a candle 1 and a spraybottle 3.
**Reasoning:** <monitor>My current sub-goal is to obtain the second soapbar. Since there are no soapbars visible here, I will attempt to examine the environment further, just in case the second soapbar is present but not immediately listed in the observation.</monitor>
**Action:** examine soapbar 2

---

**Step 15**
**Observation:** Nothing happens.
**Reasoning:** <reflection>Reflecting on my recent actions, I realize I failed to place the second soapbar in the cabinet earlier. To proceed efficiently, I need to double-check my inventory and ensure I'm actually carrying both soapbars before trying again.</reflection>
**Action:** inventory

**Step 16**
**Observation:** You are carrying: a soapbar 1.
**Reasoning:** <monitor>At this point, my sub-goal is still to locate the second soapbar. Since I am currently only carrying soapbar 1, I need to continue searching for the other one.</monitor>
**Action:** go to countertop 1

**Step 17**
**Observation:** You arrive at countertop 1. On the countertop 1, you see a soapbottle 1, a spraybottle 1, and a toiletpaper 1.
**Reasoning:** <explore>No soapbar is found on the countertop either, so I should expand my search to nearby areas. Checking places like the toilet paper hanger may help ensure I'm not missing any possible locations for the second soapbar.</explore>
**Action:** go to toiletpaperhanger 1

**Step 18**
**Observation:** On the toiletpaperhanger 1, you see a toiletpaper 3.
**Reasoning:** <explore>Since I still haven't found a new soapbar, I should continue exploring other potential areas, such as the hand towel holders, to make sure I don't overlook any possible spots where a soapbar could be located.</explore>
**Action:** go to handtowelholder 1

## H PARAMETER SENSITIVITY ANALYSIS

### H.1 ADVANTAGE COEFFICIENT $\alpha$

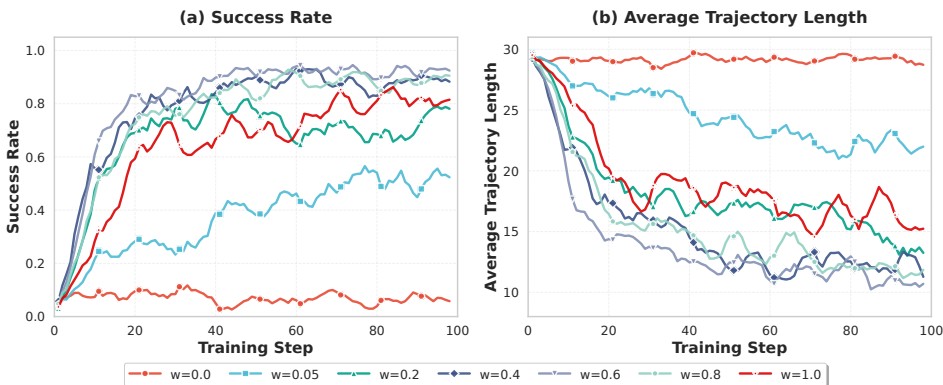

Figure 7: Sensitivity analysis of advantage coefficient $\alpha$ on ALFWorld using Qwen2.5-1.5B-Instruct.

Our composite advantage function (defined in Algorithm 2) combines two complementary signals: the outcome advantage $A_{\text{episode}}$ (which guides the agent toward task success) and the meta-reasoning advantage $A_{\text{tag}}$ (which promotes effective step-level reasoning behaviors). The hyperparameter $\alpha$ controls their relative weighting in the final advantage computation:

$$A(i) = \alpha \cdot A_{\text{episode}}(i) + (1 - \alpha) \cdot A_{\text{tag}}(i) \qquad (6)$$

To evaluate the robustness of our method to this hyperparameter choice, we performed a sensitivity analysis on ALFWorld using Qwen2.5-1.5B-Instruct with $\alpha \in \{0.0, 0.05, 0.2, 0.4, 0.6, 0.8, 1.0\}$. The results are shown in Figure 7.

- When $\alpha$ is very small (0.0 or 0.05), performance drops sharply because the local meta-reasoning reward overwhelms the global outcome signal; in fact, $\alpha = 0.0$ removes the success feedback entirely, preventing effective learning.
- When $\alpha$ is close to 1.0, the model under-weights meta-reasoning behaviors, leading to noticeably degraded success rates and longer episode lengths, which harms the quality of reasoning within successful trajectories as well as the model's generalization ability.

Empirically, performance is most stable when $\alpha$ is moderate. In this regime, outcome and meta-reasoning advantages provide complementary guidance: the model improves task success while also learning higher-quality intermediate reasoning. We also observed that, as long as $\alpha$ is not near the extremes, its effect on training speed and final performance remains small and within the natural variance of RL training. Based on these findings, and without evidence favoring a more skewed weighting, we choose $\alpha = 0.5$ as a balanced and robust setting.

### H.2 IMPACT OF DISCOUNT FACTOR

To examine whether discounting factor can further improve long-horizon learning, we introduce a standard discounted return of the form

$$G_t = \sum_{k=0}^{T-t} \gamma^k r_{t+k},$$

where $\gamma \in (0, 1]$ controls the degree of temporal discounting. Setting $\gamma = 1$ recovers the undiscounted case used in our main experiments. We conduct controlled comparisons on ALFWorld using Qwen2.5-1.5B-Instruct, evaluating both GRPO and RLVMR under $\gamma = 1$ and $\gamma < 1$.

The results in Figure 8 reveal the following trends.

For vanilla GRPO, adding a discount factor leads to a slightly faster improvement at the beginning of training, which aligns with the intuition that discounting reduces the influence of late-stage noise in sparse reward settings. However, the improvement remains modest and does not fundamentally enhance overall performance or alleviate inefficient exploration—the key limitation we target in this work.

For RLVMR, the effect of discounting is negligible. The two curves almost overlap, indicating that the explicit meta-reasoning rewards already provide dense, temporally structured guidance (e.g., discouraging repeated actions). As a result, additional temporal discounting offers little benefit on top of the meta-reasoning signals. These findings further support the robustness of RLVMR's reward design.

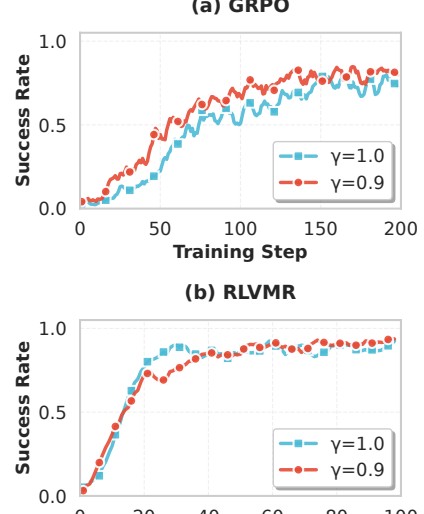

Figure 8: Comparison of methods with and without discount factor on ALFWorld using Qwen2.5-1.5B-Instruct.

## I ABLATION STUDY

### I.1 ABLATION ON KEY COMPONENTS

We conduct ablation studies on the Unseen-L2 split using Qwen2.5-1.5B-Instruct to analyze the impact of our framework's key components: the trajectory-level outcome advantage signal ($A^{\mathrm{T}}$), the meta-reasoning advantage signal ($A^{\mathrm{MC}}$), and the cold-start process (CS). The results in Table 5 confirm that each component is critical for achieving optimal performance.

Table 5: Ablation results on ALFWorld and ScienceWorld (success rates (%) on L2 variant).

| Variant | ALFWorld | ScienceWorld |
|---|---|---|
| RLVMR (Full) | **56.3** | **26.5** |
| w/o $A^{\mathrm{T}}$ (Outcome Reward) | 12.5 | 7.8 |
| w/o $A^{\mathrm{MC}}$ (Meta-Reasoning Reward) | 45.3 | 20.3 |
| w/o CS (Cold-Start) | 40.6 | 18.8 |

**Verifiable meta-reasoning rewards are essential for tackling complex, unseen tasks.** Removing the meta-reasoning advantage signal ($A^{\mathrm{MC}}$) causes a significant performance drop, with the success rate on ALFWorld falling by 11.0 percentage points (from 56.3% to 45.3%) and on ScienceWorld by 6.2 points. This variant is equivalent to a standard GRPO agent fine-tuned from the cold-start model. The sharp decline validates our central hypothesis: directly rewarding beneficial reasoning processes is crucial for developing robust problem-solving skills. This component directly addresses the "inefficient exploration issue" by providing dense, process-level signals that guide the agent toward more efficient and logical behaviors, a benefit that outcome-only rewards ($A^{\mathrm{T}}$) cannot provide alone.

**Outcome-based rewards remain indispensable for guiding the agent toward final task success.** Eliminating the trajectory-level outcome advantage ($A^{\mathrm{T}}$) results in a catastrophic performance collapse, with the success rate plummeting to just 12.5% on ALFWorld and 7.8% on ScienceWorld. This demonstrates that while meta-reasoning rewards effectively shape the **process**, the global signal of task success is vital for orienting the agent toward the ultimate goal. The meta-reasoning rewards are locally effective—for instance, rewarding non-repetitive exploration—but without the final outcome signal, the agent cannot learn which explorations ultimately lead to a successful trajectory. This confirms that the synergy between process-level and outcome-level rewards is a key strength of the RLVMR framework.

**A lightweight cold-start phase is critical for bootstrapping the agent's reasoning capabilities.** Training the agent without the supervised fine-tuning cold-start (CS) phase leads to a substantial performance decrease on both ALFWorld (down 15.7 points) and ScienceWorld (down 7.7 points). The cold-start phase, which uses only 200 trajectories, is not intended to solve the tasks but to equip the model with the basic ability to generate syntactically correct meta-reasoning tags and follow instructions. For smaller models (e.g., 1.5B), this initial grounding is vital; without it, the agent often fails to produce parsable outputs during RL, leading to training instability and policy collapse. This finding underscores the efficiency of our approach: a brief, low-data cold-start is sufficient to unlock the model's capacity for complex reasoning, which is then honed by the RL phase.

## I.2 Ablation on Meta-Reasoning Types

To further understand the role of each meta-reasoning component, we conduct a fine-grained ablation study by removing one type of meta-reasoning tag and its corresponding meta-reasoning reward at a time, while keeping all other settings unchanged. Experiments are performed on the ALFWorld L2 split using the Qwen2.5-1.5B-Instruct model.

We evaluate models using the same metrics as the main paper: (i) *success rate*, the percentage of episodes in which the agent completes the task; (ii) *average trajectory length*, the mean number of steps taken per episode, capturing overall efficiency and the agent's ability to find direct solutions; (iii) *repetitive action rate*, the percentage of actions that repeat a previous action without changing the environment state, quantifying inefficient exploration or loops; and (iv) *invalid action rate*, the proportion of actions that are not executable in the current environment state, reflecting basic comprehension and error frequency.

Table 6 summarizes the results. Removing *reflection* significantly increases both repetitive and invalid actions, indicating that the agent struggles to recover from sequences of ineffective steps without an explicit mechanism for self-correction. Removing *explore* produces substantially longer trajectories, as the agent tends to fall into inefficient search patterns without leveraging contextual cues from history to guide exploratory decisions. Eliminating either *planning* or *monitor* also leads to clear

performance degradation. We find that correct early-stage planning provides a global structure for the task, reducing disorganized execution, while monitoring helps track subgoals and maintain adherence to the planned sequence.

Overall, these ablations demonstrate that each meta-reasoning type contributes meaningfully to robust long-horizon behavior. The meta-reasoning patterns operationalized from metacognitive theory are therefore essential for improving both effectiveness and reliability of the agent.

| Variant | SR(%) ↑ | avg_steps ↓ | repeat(%) ↓ | invalid(%) ↓ |
|---|---|---|---|---|
| full RLVMR | **56.3** | **15.4** | **5.7** | **12.5** |
| w/o planning | 47.5 | 15.9 | 8.6 | 12.8 |
| w/o explore | 55.8 | 17.2 | 12.6 | 16.1 |
| w/o reflection | 46.2 | 16.5 | 14.5 | 20.2 |
| w/o monitor | 52.1 | 16.0 | 7.4 | 15.8 |

Table 6: Ablation of meta-reasoning tag types on ALFWorld L2 with Qwen2.5-1.5B-Instruct.

### I.3 IMPACT OF ANNOTATION MODEL CHOICE

To further examine the dependence of RLVMR on the teacher model used in the cold-start SFT phase, we additionally compare annotations generated by a strong closed-source model (GPT-4o) and a much smaller open-source model (Llama-3.1-70B). Specifically, we replace the cold-start annotations with those produced by different teacher models to train the Qwen2.5-1.5B-Instruct model, followed by the subsequent RL process. As shown in Figure 9, the downstream RL performance under the two annotation sources is almost identical across the training set and the most challenging L2 evaluation split for the Qwen2.5-1.5B-Instruct agents. This indicates that the annotation task required in cold-start is relatively simple, and that RLVMR's final reasoning behaviors emerge primarily from the RL stage rather than from teacher-specific annotation quality.

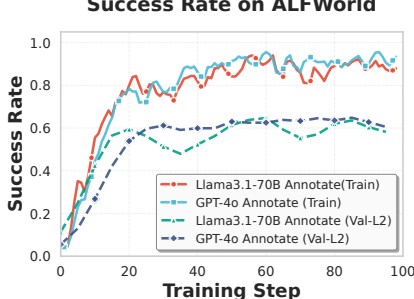

Figure 9: Success-rate on ALFWorld-L2 for Qwen2.5-1.5B-Instruct under different cold-start annotation sources.

## J THE USE OF LLMS

In this work, large language models were partially employed to assist with spelling and grammar checking, as well as minor text polishing. Specifically, we used the following prompt:

> *You are an expert in AI. Please check the provided text for any spelling or grammatical errors, and point out inappropriate expressions:* {*text segment*}

No unverifiable content was produced by the LLMs, and all technical ideas, results, and conclusions presented in this paper originate from the authors.

