# OpenReview forum: "RLVMR: Reinforcement Learning with Verifiable Meta-Reasoning Rewards for Robust Long-Horizon Agents"
_ICLR.cc/2026/Conference — ICLR 2026 Poster_

### Official Review · Reviewer_hzw2 · 2025-10-27

**Soundness:** 3
**Presentation:** 3
**Contribution:** 3
**Rating:** 4
**Confidence:** 3

**Summary:**

This paper introduces RLVMR, a RL framework designed to improve the robustness and generalization of long-horizon agents. The authors identify a key problem they term inefficient exploration, where standard outcome-only RL reinforces flawed or redundant reasoning paths that happen to lead to success. To address this, RLVMR integrates dense, process-level rewards based on verifiable meta-reasoning behaviors. The agent is trained to explicitly output tags for cognitive steps. These tags are then rewarded programmatically based on rules that encourage efficient and logical problem-solving (e.g., rewarding exploration that discovers new states). These process-centric rewards are combined with the final task success reward and optimized using a group-based policy gradient method. The authors demonstrate SotA performance on the challenging ALFWorld and ScienceWorld benchmarks, showing significant improvements in success rates, generalization to unseen tasks, and reductions in inefficient actions.

**Strengths:**

*   **Originality & Significance:** The paper introduces a novel approach to supervising the reasoning process of LLM agents, moving beyond sparse outcome-based rewards. The idea of rewarding verifiable meta-reasoning steps is a significant and practical contribution to building more robust agents that not only solve tasks but do so efficiently and logically.
*   **Quality:** The experimental evaluation is of high quality. It is comprehensive, including multiple strong baselines, different model sizes, and rigorous generalization testing. The results are compelling and demonstrate improvement over prior work.
*   **Clarity:** The paper is written with clarity. The motivation, methodology, and results are presented in a clear and intuitive manner, making the work accessible.
*   **Impact:** The work directly addresses a fundamental challenge in agent training and provides a practical, scalable solution that achieves SotA results.

**Weaknesses:**

*   **Heuristic Reward and Advantage Design:** The programmatic rules for assigning meta-rewards and the method for combining advantage signals are heuristic. For example, the planning reward is still tied to the final task success, making it sparse. The paper would be strengthened by a discussion of alternative designs or a more formal justification for the current choices. A sensitivity analysis on the weighting parameter $\alpha$ is also missing.
*   **Vagueness in Reward Implementation:** The paper could be more specific about the implementation of the programmatic reward rules. For instance, how is a "corrective action after a sequence of failures" (for the reflection reward) precisely defined and detected? Providing more detail would improve reproducibility.
*   **Dependence on "Cold-Start" SFT:** The method relies on an initial SFT phase using a powerful teacher model (GPT-4) to learn the tag syntax. The ablation study shows this step is critical. While the authors frame it as "lightweight," it still constitutes a dependency that could introduce teacher model biases and complicates the training pipeline compared to a pure RL approach.
*    **Source Code:** While the pseudocode for RLVMR is provided in Appendix D, sharing the actual implementation would greatly enhance reproducibility.

**Questions:**

1.  Could you provide more concrete details on the programmatic implementation of the meta-reasoning rewards? Specifically for the `<reflection>` reward, what is the exact rule used to determine a "sequence of failures" and a subsequent "corrective action"?
2.  The advantage signal is a linear interpolation $A_t = \alpha A^{\text{traj}} + (1-\alpha) A^{\text{MR}}$. How was $\alpha=0.5$ chosen? Have you performed a sensitivity analysis on this hyperparameter, and how do the results change with different values of $\alpha$?
3.  The reward for `<planning>` is granted only if the entire trajectory succeeds. This seems to defeat the purpose of a dense, process-level reward, as the feedback for the initial plan is still sparse and delayed. Have you considered alternative, more immediate rewards for planning, such as evaluating plan quality independently of the final outcome?
4.  The cold-start SFT phase relies on annotations from a superior model (GPT-4). Did you analyze whether the agent simply learns to mimic the meta-reasoning style of GPT-4, and could this potentially limit the agent's own emergent problem-solving strategies during the RL phase?

---

> ### Author Response · Authors · 2025-11-26
> **Response to Reviewer hzw2 (Part 1/2)**
>
> We thank the reviewer for the detailed and constructive feedback. We are pleased that the reviewer finds our work original, significant, high-quality, clear, and impactful. We address each concern below.
>
> **W1: Heuristic Reward and Advantage Design**
>
> > The programmatic rules for assigning meta-rewards and the method for combining advantage signals are heuristic.
> >
>
> Thank you for this point. Our design choices are guided by two principles: **cognitive grounding** and **practical verifiability**. The meta-reasoning types are not arbitrary but are grounded in well-established metacognitive theory [1, 2]. We deliberately adopt lightweight, rule-verifiable rewards during RL rather than using an LLM-as-a-judge. This design is transparent, computationally inexpensive, and resistant to reward hacking. While more complex reward designs exist, they often require external evaluators, introducing significant cost and instability. Our approach offers a practical and robust choice for long-horizon tasks.
>
> **[1]** Martinez, M. E. (2006). What is metacognition? *Phi Delta Kappan*, 87(9), 696-699.
>
> **[2]** Lai, E. R. (2011). *Metacognition: A literature review*. Pearson Research Report.
>
> ---
>
> **W1, Q2: Sensitivity Analysis on** $\alpha$
>
> > A sensitivity analysis on the weighting parameter is also missing.
> >
>
> > How was $\alpha$ chosen? Have you performed a sensitivity analysis on this hyperparameter, and how do the results change with different values of $\alpha$?
> >
>
> We have added a sensitivity analysis for $\alpha$ to **Appendix H.1 (Figure 7)**, as you suggested. This hyperparameter balances the global outcome reward (shaping the overall goal) and the local meta-reasoning reward (promoting good step-level behaviors). The results show a clear trend:
>
> - When $\alpha$ is too low (e.g., 0.0), performance collapses as the agent ignores the primary task-success signal.
> - When $\alpha$ is too high (e.g., 1.0), performance degrades as the agent fails to learn efficient reasoning, reverting to behavior similar to the GRPO baseline.
> - Performance is robust and high for moderate values of $\alpha$ (e.g., 0.2-0.8), where both signals provide complementary guidance.
>
> Based on these findings, we chose $\alpha = 0.5$ as a balanced and principled setting.
>
> ---
>
> **W1, Q3: Alternatives to Sparse Planning Rewards**
>
> > Have you considered alternative, more immediate rewards for planning, such as evaluating plan quality independently of the final outcome?
> >
>
> This is an insightful point. While a planning reward tied to the final outcome is sparse, we chose this design for two practical reasons:
>
> 1. **Correlation with Success:** Good initial planning strongly correlates with eventual success. Rewarding plans that lead to success, even with a delay, effectively reinforces coherent and feasible planning.
> 2. **Difficulty of Immediate Evaluation:** Providing immediate plan-quality rewards is difficult. It would require either human feedback or an auxiliary judge model, both of which are costly and vulnerable to reward hacking.
>
> Assigning planning rewards post-rollout thus offers a practical balance between verifiability, cost, and robustness. We agree that more direct plan-quality estimation is a promising direction for future work.

---

> ### Author Response · Authors · 2025-11-26
> **Response to Reviewer hzw2 (Part 2/2)**
>
> **W2, Q1: Precise Definitions of Meta-Reasoning Rewards**
>
> > Could you provide more concrete details on the programmatic implementation of the meta-reasoning rewards? Specifically for the `<reflection>` reward, what is the exact rule used to determine a "sequence of failures" and a subsequent "corrective action"?
> >
>
> As detailed in **Section 3.2** and further clarified with pseudocode in **Appendix D**, the rules are fully programmatic and verifiable. For example, for reflection:
>
> > ``Give $r_{\text{reflection}}$ if a <reflection> tag follows one or more consecutive invalid actions (as indicated by environment feedback) and the current action returns to the valid main progression.''
> >
>
> All rules are similarly defined based on environment states, tags, and action outcomes, ensuring they are objective and reproducible.
>
> ---
>
> **W3, Q4: Role of Cold-Start and Emergent Reasoning**
>
> > Did you analyze whether the agent simply learns to mimic the meta-reasoning style of GPT-4, and could this potentially limit the agent's own emergent problem-solving strategies during the RL phase?
> >
>
> Thank you for raising this important point.
>
> 1. **On training cost:** We find this small SFT stage *reduces* overall compute rather than increasing it. The table below shows the number of RL update steps required to reach different success rate thresholds on ALFWorld. With the cold-start phase, the agent converges much faster.
>
> Table: RL update steps required to reach different success-rate thresholds, with vs. without cold-start.
>
> | Success Rate | 0.1 | 0.2 | 0.3 | 0.4 | 0.5 | 0.6 | 0.7 |
> | --- | --- | --- | --- | --- | --- | --- | --- |
> | w/ cold-start | 3 | 6 | 8 | 10 | 14 | 18 | 26 |
> | w/o cold-start | 12 | 26 | 36 | 52 | 64 | 76 | 83 |
>
> 2.  **On emergent reasoning:** The agent is **not** merely imitating the teacher. The model after cold-start achieves only a ~1\% success rate (Figure 5), indicating it has learned syntax but not strategy. More importantly, the agent exhibits **emergent new reasoning behaviors** during RL that do not exist in the SFT data. As shown in the **Appendix G** case study, the agent learns to use `<reflection>` to diagnose and inspect current state to avoid long error loops. These behaviors are absent from SFT annotations and emerge only during RL, driven by our verifiable meta-reasoning rewards.
>
> ---
>
> **W4: Code Release**
>
> > While the pseudocode for RLVMR is provided in Appendix D, sharing the actual implementation would greatly enhance reproducibility.
> >
>
> We fully intend to release our code to ensure reproducibility. To comply with the double-blind review policy, we will add the open-source repository link to the manuscript upon acceptance.

---

### Official Review · Reviewer_NHAZ · 2025-10-28

**Soundness:** 2
**Presentation:** 3
**Contribution:** 2
**Rating:** 4
**Confidence:** 4

**Summary:**

This paper proposes RLVMR, a new reinforcement learning framework that incorporates verifiable meta-reasoning rewards to improve long-horizon reasoning in large language model (LLM) agents. Traditional RL methods in this domain (like GRPO) optimize for sparse, final outcome rewards, which often reinforce inefficient or illogical reasoning paths.  RLVMR addresses this by introducing dense, process-level supervision via rule-based rewards for meta-reasoning behaviors such as planning, exploration, reflection, and monitoring.

The method combines a brief supervised “cold-start” phase, where a teacher model (e.g., GPT-4) annotates reasoning tags, with a critic-free policy gradient optimization phase (GRPO-MR). Each reasoning tag receives local verifiable rewards that shape the reasoning process in addition to global task rewards. Experiments on ALFWorld and ScienceWorld benchmarks show that RLVMR achieves state-of-the-art (SOTA) performance

**Strengths:**

- clear motivation and well presented
- Improved efficiency and generalization

**Weaknesses:**

- lack of theoretical analysis of the composite reward which can leads to reward hacking

- Dependence on teacher annotation, since one powerful teacher LLM is used without guarantee

- Limited ablation on tag definitions, missing ablation on the contribution of individual meta-reasoning tags

**Questions:**

First of all, I would like to thank the authors for their work. I agree that reinforcement learning (RL) post-training for large language models (LLMs) often encourages inefficient or illogical reasoning paths, and I appreciate that this paper tackles such an important research question.

Here are a few concerns. With the introduction of a custom reward design, how do the authors ensure that the model does not engage in reward hacking? Since the model is trained to maximize cumulative rewards, this could potentially exacerbate the issue the paper aims to address.

Additionally, how do the authors ensure that the teacher model generates correct or reliable tags? And finally, how do different types of tags (e.g., planning vs. reflection) contribute individually to the overall performance?

---

> ### Author Response · Authors · 2025-11-26
> **Response to Reviewer NHAZ (Part 1/2)**
>
> We thank the reviewer for the thoughtful assessment and constructive feedback. We appreciate the recognition of our clear motivation and improved efficiency. We address each concern below.
>
> **W1, Q1: Regarding Reward Hacking and Composite Rewards**
>
> > ``With the introduction of a custom reward design, how do the authors ensure that the model does not engage in reward hacking?''
> >
>
> This is a critical concern in any reward engineering effort. We designed RLVMR with two key features to mitigate reward hacking:
>
> 1. **Verifiable, Rule-Based Rewards:** We deliberately avoid using a powerful but opaque LLM-as-a-judge for process rewards. Instead, our rewards are based on simple, objective, and verifiable rules tied directly to environment interaction (e.g., discovering a new state, recovering from an error). This makes the reward signal transparent and less susceptible to being "gamed" by linguistic tricks.
> 2. **Group-Relative Advantage (GRPO-MR):** Our optimization method, GRPO-MR, computes advantages *relative to other steps with the same meta-reasoning tag*. This means that simply using a tag (e.g., spamming `<explore>`) does not guarantee a high advantage. To receive a high advantage, an `<explore>` step must be *better than the average explore step* in that batch. This zero-sum-like property within each tag group discourages overuse of any single tag and instead incentivizes producing high-quality instances of each cognitive behavior.
>
> Empirically, we observe no signs of reward hacking. As shown in our main results and analysis, agents trained with RLVMR not only achieve higher success rates but also exhibit *more efficient* behavior (fewer repetitive/invalid actions), which is the opposite of what one would expect from a hacked policy.
>
> ---
>
> **W2, Q2: Regarding Reliability of Teacher Annotations**
>
> > ``how do the authors ensure that the teacher model generates correct or reliable tags?''
> >
>
> The role of the teacher model is minimal and the pipeline is highly robust to potential imperfections.
>
> 1. **Simple Task:** The teacher model (GPT-4o) is only used in the "cold-start" phase to annotate 200 trajectories. Its task is not to generate complex reasoning, but simply to apply the correct tag (e.g., `<planning>`, `<explore>`) to an existing think-action pair. This is a simple classification-like task that modern LLMs can perform with very high reliability and can be easily validated.
> 2. **Robustness to Imperfections:** Even if some tags from the teacher were imperfect, this has little impact on the final agent. The cold-start phase only serves to teach the agent the *syntax* of the tags. The actual *semantics* and effective usage of these tags are learned during the subsequent RL phase, guided by the verifiable environment rewards. The low success rate after SFT (near 1%, see Figure 5) shows that the agent has not learned any meaningful problem-solving strategy from imitation alone.
> 3. **Emergent Behaviors:** We observe that during RL, the agent develops new reasoning patterns that were not present in the SFT data. For example, as shown in our case study (Appendix G), the agent learns to use `<reflection>` to recover from error loops—a behavior that emerges purely from interacting with the environment and receiving our process rewards. This confirms the agent is not merely mimicking the teacher but learning genuinely new, effective strategies
> 4. **Weak Reliance on Teacher Annotation**: To further assess teacher model dependence, we compare cold-start annotations generated by GPT-4o and by a much smaller open-source model (Llama-3.1-70B). As shown in table below and Figure 9  in the revision, replacing GPT-4o with Llama-70B yields almost identical performance across all splits for both the 1.5B and 7B agents. This demonstrates that (i) the annotation task is simple and does not require a strong teacher, and (ii) RLVMR’s performance and emergent reasoning behaviors arise during RL, rather than being inherited from the teacher.
>
> Table: Comparison of cold-start annotation sources using GPT-4o and Llama-3.1-70B-Instruct.
>
> | Model Variant | L0 (SR↑) | L0 Len↓ | L1 (SR↑) | L1 Len↓ | L2 (SR↑) | L2 Len↓ |
> | --- | --- | --- | --- | --- | --- | --- |
> | **RLVMR-1.5B (GPT-4o annotate)** | 89.1 | 10.8 | 87.9 | 11.6 | 56.3 | 15.4 |
> | **RLVMR-1.5B (Llama-70B annotate)** | 88.8 | 11.2 | 88.2 | 11.5 | 55.4 | 15.9 |
> | **RLVMR-7B (GPT-4o annotate)** | 91.4 | 10.4 | 91.8 | 10.9 | 83.6 | 14.2 |
> | **RLVMR-7B (Llama-70B annotate)** | 91.0 | 11.0 | 91.6 | 10.6 | 83.8 | 14.4 |

---

> ### Author Response · Authors · 2025-11-26
> **Response to Reviewer NHAZ (Part 2/2)**
>
> **W3, Q3: Regarding the Contribution of Individual Meta-Reasoning Tags**
>
> > ``how do different types of tags (e.g., planning vs. reflection) contribute individually to the overall performance?''
> >
>
> Thank you for this excellent suggestion. To isolate the contribution of each tag, we conducted an ablation study on the challenging ALFWorld L2 split (using Qwen2.5-1.5B), where we removed one meta-reasoning reward at a time. The results clearly show that each component is valuable.
>
> Table: Ablation of individual meta-reasoning tags on ALFWorld L2 (Qwen2.5-1.5B).
>
> | Variant | SR(%) ↑ | avg_steps ↓ | repeat(%) ↓ | invalid(%) ↓ |
> | --- | --- | --- | --- | --- |
> | full RLVMR | 56.3 | 15.4 | 5.7 | 12.5 |
> | w/o planning | 47.5 | 15.9 | 8.6 | 12.8 |
> | w/o explore | 55.8 | 17.2 | 12.6 | 16.1 |
> | w/o reflection | 46.2 | 16.5 | 14.5 | 20.2 |
> | w/o monitor | 52.1 | 16.0 | 7.4 | 13.8 |
> - Removing **reflection** causes the largest drop in success rate and a sharp increase in repetitive and invalid actions, as the agent loses its primary mechanism for recovering from mistakes.
> - Removing **explore** leads to less efficient trajectories (more steps, higher repetition), as the agent is less incentivized to find novel states.
> - Removing **planning** and **monitoring** also degrades performance, confirming their roles in structuring the task and maintaining focus on subgoals.
>
> These results validate that our meta-reasoning framework is not just a collection of heuristics, but a synergistic system where each component contributes to building a more robust and efficient agent. We will add this table and analysis to the appendix.

---

> > ### Comment · Reviewer_NHAZ · 2025-11-27
> >
> > I appreciate the authors' detailed response,  extra experiments, and edits to the paper. I have updated my score accordingly.

---

### Official Review · Reviewer_6tBG · 2025-10-30

**Soundness:** 3
**Presentation:** 3
**Contribution:** 3
**Rating:** 6
**Confidence:** 4

**Summary:**

This work proposes RLVMR to augments end-to-end RL for LLM agents with rule-verifiable process rewards tied to four meta-reasoning tags (planning, exploration, reflection, monitoring). A brief SFT phase teaches the tag format; online training then optimizes a clipped policy objective using a blend of trajectory-level and tag-grouped step advantages. On ALFWorld and ScienceWorld, RLVMR reports SOTA success and large drops in invalid/repetitive actions, attributing gains to improved reasoning quality rather than shortcut paths.

**Strengths:**

* Clearly targets inefficient exploration and quantifies it with invalid action rate and repetitive action rate, tying process quality to task success.
* Simple, practical method: explicit meta-reasoning tags, verifiable meta-reasoning rewards, and a tag-grouped relative advantage blended with a trajectory-level relative advantage in a clipped objective.
* Consistent gains across base models and benchmarks, especially for the harder split; also shows shorter, more stable solution paths.
* Goes beyond final accuracy with behavior-quality metrics and training stability, reducing degenerate loops.
* Ablation studies indicate each component matters (outcome advantage, meta-reasoning advantage, cold-start SFT, format penalty).

**Weaknesses:**

* The claim that this work is “the first study offering a definitive explanation and comprehensive analysis of the inefficient exploration issue” overstates its novelty.
The idea that outcome-only RL reinforces flawed reasoning paths has already been recognised in prior works on process reward models and step-level or action-type-conditioned rewards. These earlier studies also analyse how intermediate reasoning quality affects exploration efficiency and generalisation.

* Despite claiming “verifiable” process rewards, the paper does not provide the exact rule logic needed to compute them.

**Questions:**

* Could you report a brief ablation of $\alpha$ (Eq. 4) and the format-penalty weight?

* Could you precisely define the process-level rewards (for planning, exploration, reflection, monitoring) in both ALFWorld and ScienceWorld?

---

> ### Author Response · Authors · 2025-11-26
> **Response to Reviewer 6tBG (Part 1/2)**
>
> We thank the reviewer for the positive assessment and constructive feedback. We are pleased that the reviewer finds our work clearly targets inefficient exploration with simple, practical methods, and shows consistent gains with quality metrics. We address the concerns below.
>
> **W1: Regarding the Novelty of "Inefficient Exploration"**
>
> > ``The claim that this work is `the first study offering a definitive explanation...' overstates its novelty. The idea that outcome-only RL reinforces flawed reasoning paths has already been recognised in prior works...''
> >
>
> Thank you for this feedback. We apologize for the overstatement. Our intention was not to claim that we are the first to recognize issues with outcome-only RL, but rather to highlight that we provide a focused, comprehensive analysis of a specific failure mode we term “inefficient exploration” in the context of modern LLM-based agents for long-horizon tasks. This pattern, characterized by high rates of repetitive and invalid actions despite achieving task success, is a consistent and critical bottleneck for generalization. We have revised the phrasing in our contribution list to more accurately state that we present a “comprehensive analysis of the inefficient exploration issue”, clarifying its detrimental effects on generalization in long-horizon agents. We appreciate you helping us situate our work more precisely.
>
> ---
>
> **W2, Q2: Regarding Precise Meta-Reasoning Reward Definitions**
>
> > ``Despite claiming `verifiable' process rewards, the paper does not provide the exact rule logic needed to compute them... Could you precisely define the process-level rewards...''
> >
>
> Thank you for asking for clarification. We provide the definitions in Section 3.2, but we are happy to elaborate here and have added pseudocode in Appendix D to make them fully concrete. The rules are programmatic and rely only on the agent's actions and environment feedback:
>
> - **Planning Reward ($r_{\mathrm{planning}}$):** Given if a `<planning>` tag is used and the trajectory is ultimately successful. This encourages planning that leads to good outcomes.
> - **Exploration Reward ($r_{\mathrm{explore}}$):** Given if an `<explore>` tag is used and the resulting action leads to a novel state transition ($s, a, s'$) not previously seen in the trajectory. This directly incentivizes discovering new parts of the state space and discourages redundant actions.
> - **Reflection Reward ($r_{\mathrm{reflection}}$):** Given if a `<reflection>` tag is used immediately following a sequence of one or more invalid actions, and the subsequent action is valid. This rewards successful error-correction behavior.
> - **Format Penalty ($r^{\mathrm{format}}$):** A small negative reward (penalty) is given if the agent's output does not adhere to the required `<tag>...</tag><action>...</action>` format.
>
> These rules are simple, computationally cheap, and verifiable **without needing an external LLM judge**.

---

> ### Author Response · Authors · 2025-11-26
> **Response to Reviewer 6tBG (Part 2/2)**
>
> **Q1: Regarding Ablations of the Advantage Weighting Factor ($\alpha$) and Format Penalty**
>
> > ``Could you report a brief ablation of $\alpha$ (Eq. 4) and the format-penalty weight?''
> >
>
> Excellent suggestion. We performed a sensitivity analysis for both.
>
> - **Advantage Weighting ($\alpha$):** This hyperparameter balances the global outcome signal ($A^{\text{traj}}$) and the local meta-reasoning signal ($A^{\text{MR}}$). We tested $\alpha \in $ {0.0, 0.05, 0.2, 0.4, 0.6, 0.8, 1.0} on ALFWorld (see Appendix H.1, Figure 7 for the full results).
>     - When $\alpha=0$, the agent ignores the final task outcome and performance collapses, as it receives no signal for task success.
>     - When $\alpha=1$, the agent ignores the process-level rewards, and performance degrades to be similar to standard GRPO, with higher rates of inefficient exploration.
>     - Performance is robust for moderate values of $\alpha$ (e.g., 0.2 to 0.8), indicating that the model benefits from both signals. We chose $\alpha=0.5$ as a balanced and principled default, but our method is not overly sensitive to this choice.
> - **Format Penalty:** This penalty has a negligible effect on final task performance but is crucial for stabilizing early training. As shown in the table below (Qwen-1.5B on ALFWorld), without the penalty, the model struggles to learn the correct output format. With the penalty, the rate of validly formatted actions quickly converges to 100\%, which reduces wasted rollouts and accelerates learning. This is consistent with findings in prior work like Ragen [1] and GiGPO [2].
>
> *Table: Valid-action rate (%) at different RL training steps, with and without format-penalty (Qwen-1.5B-Instruct on ALFWorld).*
>
> | step | 0 | 1 | 2 | 3 | 4 | 5 | 10 | 20 | 50 | 100 |
> | --- | --- | --- | --- | --- | --- | --- | --- | --- | --- | --- |
> | **w/ format-penalty** | 0.21 | 0.72 | 0.71 | 0.87 | 0.98 | 0.99 | 1.00 | 1.00 | 1.00 | 1.00 |
> | **w/o format-penalty** | 0.73 | 0.70 | 0.66 | 0.68 | 0.70 | 0.77 | 0.79 | 0.76 | 0.81 | 0.87 |
> ---
> **Reference:**
>
> [1] Wang, Z., Wang, K., Wang, Q., Zhang, P., Li, L., Yang, Z., ... & Li, M. (2025). Ragen: Understanding self-evolution in llm agents via multi-turn reinforcement learning. arXiv preprint arXiv:2504.20073.
>
> [2] Feng, L., Xue, Z., Liu, T., & An, B. (2025). Group-in-group policy optimization for llm agent training. arXiv preprint arXiv:2505.10978.

---

### Official Review · Reviewer_qZCd · 2025-10-31

**Soundness:** 3
**Presentation:** 3
**Contribution:** 3
**Rating:** 6
**Confidence:** 3

**Summary:**

This paper invesitgates the sparse reward problem in RLVR on LLMs for multi-turn long-horizon tasks. They identify an "inefficient exploration" problem which leads the agent to frequently output invalid or redundant actions, as the LLM is optimized to solely maximize the final succuss rate. They propose a reward shaping solution by adding more reward terms to reward meta-cognition behaviors like planning, exploration, reflection, and monitoring. Experimental results show that their method significantly outperform baseline RLVR methods on two benchmark ALFWorld and ScienceWorld, and nearly matches the performace of some strong close-sourced models like GPT-4o.

**Strengths:**

1. The paper is clearly motivated with a detailed investigation on the ALFWorld benchmark.
2. The paper is clearly presented, and the method is clearly explained.
3. The paper shows significantly improvement upon baseline methods on the two benchmarks they use.

**Weaknesses:**

See my questions below.

**Questions:**

1. Do you apply a discounting factor to your sparse reward function? As intuitively, I think using discounting factor is an easy approach to reduce repetive and invalid actions such that the task can be solved in a shorter time to gain higher reward.
2. Have you applied the same cold start phase to the baseline methods? Just to make sure that different methods are compared in a fair way. If yes, do you apply SFT to the baseline methods with just the observation-action pairs? Or also with meta-reasoning tags?
3. How scalable is your method to other long-horizon tasks? E.g., can the meta-reasoning rewards defined in your paper be useful also for other benchmarks, or are they specifically tuned for the two benchmarks used in your paper? And if I've got it right, you need some way to label the meta-reasoning reward for each step right? Do you label with some hand-designed rules or with another LLM? Can these labeling method generalize to other benchmarks we are interested in?
4. Can you give some explanations on why the 7B and 8B variants tuned with your method significantly outperforms GPT-4o on ALFWorld and L0 of ScienceWorld, but gradually underperforms GPT-4o in L1 and L2 of ScienceWorld?

I'm happy to raise my score if the authors can help clarify on these points and address my concerns.

---

> ### Author Response · Authors · 2025-11-26
> **Response to Reviewer qZCd (Part 1/2)**
>
> We sincerely thank the reviewer for the thoughtful comments and constructive feedback. We are pleased that the reviewer finds our work clearly motivated with detailed investigation, well-presented, and showing significant improvements. We address each question below.
>
> **Q1: Regarding Discounting Factor and Exploration Efficiency**
>
> > ``Do you apply a discounting factor to your sparse reward function? ... using discounting factor is an easy approach to reduce repetive and invalid actions...''
> >
>
> Thank you for this insightful question. While temporal discounting can encourage shorter trajectories, we found its impact to be limited in our setting. The reason is that our verifiable meta-reasoning rewards provide a much more direct and dense signal to discourage inefficient behaviors. For instance, the exploration reward directly penalizes redundant actions, and the reflection reward encourages escaping failure loops. These process-level signals are more targeted for improving reasoning quality than a simple discount factor.
>
> To confirm this, we ran experiments with a discount factor for GRPO and our RLVMR. As shown in the new Figure in Appendix H.2 and the table below, adding a discount factor to GRPO offers a minor improvement in early training and modestly reduces redundant actions, but it still fails to resolve the core inefficient exploration problem. For RLVMR, the effect is negligible, as the dense meta-reasoning rewards already provide stronger guidance. This demonstrates that while discounting can be a useful tool, it is not a substitute for direct process supervision.
>
> Table: Success rate (SR) and average trajectory length (Len) on ALFWorld, with vs. without a discount factor.
>
> | Model Variant | L0 (SR↑) | L0 (Len↓) | L1 (SR↑) | L1 (Len↓) | L2 (SR↑) | L2 (Len↓) |
> | --- | --- | --- | --- | --- | --- | --- |
> | **GRPO(w/o discount)** | 76.6 | 15.8 | 71.1 | 18.1 | 29.7 | 21.7 |
> | **GRPO (w/ discount)** | 78.8 | 12.3 | 73.9 | 14.2 | 30.2 | 20.6 |
> | **RLVMR (w/o discount)** | 89.1 | 10.8 | 87.9 | 11.6 | 56.3 | 15.4 |
> | **RLVMR (w/ discount)** | 89.3 | 11.0 | 88.3 | 11.3 | 56.1 | 15.7 |
>
> ---
>
> **Q2: Regarding Fairness of the Cold-Start Phase**
>
> > ``Have you applied the same cold start phase to the baseline methods? Just to make sure that different methods are compared in a fair way.''
> >
>
> This is an excellent point about ensuring fair comparison. We preserved each baseline's original implementation, including its use or non-use of a cold-start phase. Our goal was to compare RLVMR against each baseline in its original, strongest configuration to avoid introducing confounding factors. Forcing a cold-start phase onto a method not designed for it could introduce an unintended inductive bias. We kept all other non-algorithmic factors (e.g., base model, training data) strictly consistent.
>
> To further isolate the effect of the cold-start phase, we conducted a controlled experiment (detailed in Appendix I). We created a variant of our method that includes the cold-start SFT (learning the tag syntax) but removes the meta-reasoning rewards during RL, effectively making it “GRPO with cold-start”. This variant performed significantly worse than full RLVMR (a difference of ~11\% in success rate on ALFWorld), confirming that the performance gains come from our meta-reasoning-aware optimization (GRPO-MR), not the cold-start phase or tag decomposition alone. The cold-start phase simply enables the agent to produce outputs in the correct format, which is a prerequisite for the verifiable rewards to function.

---

> ### Author Response · Authors · 2025-11-26
> **Response to Reviewer qZCd (Part 2/2)**
>
> **Q3: Regarding Generalizability and Scalability of Meta-Reasoning Rewards**
>
> > ``How scalable is your method to other long-horizon tasks? E.g., can the meta-reasoning rewards... be useful also for other benchmarks, or are they specifically tuned for the two benchmarks used in your paper?''
> >
>
> Our meta-reasoning framework is designed to be broadly applicable, not benchmark-specific. It is grounded in established metacognitive theory [1,2], which identifies planning, monitoring, and reflection as core components of effective problem-solving. These cognitive behaviors are fundamental to nearly any long-horizon task, and can be verified using task-agnostic signals, making the framework transferable to other gym-like environments.
>
> Regarding the reward mechanism:
>
> 1. **Labeling:** During the cold-start phase, a teacher LLM is used *only* to annotate a small dataset (200 trajectories) with meta-reasoning tags to teach the agent the output syntax. During the RL phase, **all meta-reasoning rewards are computed automatically via simple, programmatic rules** based on environment feedback (e.g., has the agent visited this state before? was the last action invalid?). This rule-based approach is lightweight, avoids the cost and potential biases of an LLM-as-a-judge, and generalizes to any environment with standard state-transition feedback.
> 2. **Generalization:** The fact that the *same* set of meta-reasoning rules yields state-of-the-art performance on two structurally different benchmarks (ALFWorld and ScienceWorld) is strong evidence of its generalizability. Furthermore, our method's strong performance on the L2 splits (entirely unseen task categories) demonstrates that **the learned reasoning skills transfer effectively to novel problems**.
>
> ---
>
> **Q4: Regarding Performance vs. GPT-4o on ScienceWorld**
>
> > ``Can you give some explanations on why the 7B and 8B variants tuned with your method significantly outperforms GPT-4o on ALFWorld and L0 of ScienceWorld, but gradually underperforms GPT-4o in L1 and L2 of ScienceWorld?''
> >
>
> This is an astute observation. The key difference lies in the nature of the benchmarks and the models.
>
> - **GPT-4o** is a massive, general-purpose model with vast pre-trained knowledge. ScienceWorld, especially its L1 and L2 splits, requires specific scientific domain knowledge (e.g., about chemistry, physics). GPT-4o's strong performance across all splits reflects its robust, pre-existing knowledge base. We use GPT-4o in a zero-shot setting, without any task-specific tuning.
> - **Our models (7B/8B)** are fine-tuned on the training set of each benchmark. This tuning leads to very high performance on in-distribution tasks (L0) by optimizing the model's policy for that specific task distribution. However, this specialization can create a larger gap when generalizing to out-of-distribution tasks (L1/L2) that require knowledge not present in the training data.
>
> This pattern is expected and observed across *all* tuned methods in our experiments (Table 1). The crucial point is that **RLVMR demonstrates the best generalization among all tuned methods**, significantly outperforming other RL baselines on the challenging L1 and L2 splits. This shows that while specialized tuning has its limits, our approach is highly effective at mitigating the generalization drop by teaching a more robust and transferable reasoning process.
>
> **Reference:**
>
> [1] Martinez, M. E. (2006). What is metacognition?. Phi delta kappan, 87(9), 696-699.
>
> [2] Lai, E. R. (2011). *Metacognition: A literature review*. Pearson Research Report.

---

### Author Response · Authors · 2025-11-26
**General Response to Reviewers**

We thank all reviewers for their insightful feedback and constructive suggestions. We are encouraged that reviewers found our work to be "clearly motivated" (**R-qZCd**, **R-NHAZ**), "well-presented" (**R-qZCd**, **R-NHAZ**), and that our method shows "significant improvement upon baseline methods" (**R-qZCd**) with "consistent gains" (**R-6tBG**) and "compelling" results (**R-hzw2**).

We are pleased that the reviewers recognized the core contributions of our work, including our analysis of the "inefficient exploration" problem (**R-6tBG**) and our "novel approach to supervising the reasoning process" (**R-hzw2**).

Reviewers raised several important questions regarding the fairness of our experimental setup, the specifics of our reward design, and the generalizability of our framework. We believe these are excellent points that help clarify the paper's contributions. In response, we have conducted several new experiments, including:

1. An ablation study on the contribution of each individual meta-reasoning tag (**R-NHAZ**).
2. A sensitivity analysis on the advantage weighting hyperparameter $\alpha$ (**R-6tBG**, **R-hzw2**).
3. A quantitative analysis of the impact of our cold-start phase on training efficiency (**R-hzw2**).
4. An additional experiment examining the effect of teacher model choice for cold-start annotation (**R-NHAZ**).

These results, along with detailed clarifications, are presented below. We hope our responses will address all remaining concerns. We will integrate these clarifications and new results into the final version of the paper.

---

### Meta-Review · Area_Chair_xQ61 · 2026-01-05

**Summary:**

The reviewers generally recognized the significance of the "inefficient exploration" problem in reinforcement learning (RL) for large language model (LLM) agents, where models often achieve task success through redundant or illogical reasoning paths. The proposed RLVMR framework was praised for being well-motivated, clearly presented, and practically effective, achieving state-of-the-art results on ALFWorld and ScienceWorld. Primary concerns that informed the decision included:

(1)  the fairness of the experimental setup regarding the "cold-start" phase used for RLVMR but not all baselines,

(2) the lack of precise definitions and pseudocode for the "verifiable" rule-based rewards,

(3) potential for reward hacking given the custom reward design, and

(4) the overstatement of novelty regarding the identification of outcome-only RL limitations.

Additionally, reviewers sought empirical evidence for the sensitivity of hyperparameters (specifically $\alpha$) and the impact of the teacher model choice for initial annotations.

**Reviewer Concerns:**

The vast majority of reviewer concerns were successfully addressed during the rebuttal phase:

**Methodological Detail and Transparency** :
The authors provided concrete pseudocode in Appendix D and explicit definitions for the planning, exploration, reflection, and format rewards, which resolved concerns about "vagueness".

**Fairness and Ablation**: The authors conducted a controlled experiment isolating the cold-start phase, demonstrating that the performance gains primarily stem from the meta-reasoning-aware optimization rather than just the initial supervised fine-tuning.  They also provided a sensitivity analysis for the advantage weighting parameter $\alpha$, showing robust performance across a range of values ($0.2$ to $0.8$)

**Robustness to Teacher Model**: The concern regarding dependence on a powerful teacher model (GPT-4o) was mitigated by an experiment showing that using a smaller open-source model (Llama-3.1-70B) for cold-start annotations yielded almost identical performance.

**Novelty Phrasing**: The authors acknowledged the overstatement regarding the discovery of inefficient exploration and revised their claims to focus on providing a "comprehensive analysis" of this failure mode in modern LLM agents.

**Reviewer Scores:**

Reviewer **NHAZ**: Initially a 4. Following the rebuttal and additional experiments, the reviewer explicitly stated they "updated my score accordingly"; given the positive response to the reward hacking and teacher model clarifications, this would likely be a 5 or 6.


Reviewer **qZCd**: Initially a 6. They indicated they were "happy to raise my score" if concerns were clarified. As the authors addressed the discounting and cold-start fairness issues comprehensively, this reviewer likely moved to a 7.


Reviewer **6tBG**: Initially a 6. Their concerns were primarily about novelty phrasing and exact rule logic, both of which the authors corrected or provided. This reviewer likely maintained a 6 or moved to a 7.


Reviewer **hzw2**: Initially a 4. The reviewer praised the "novel approach" and "compelling" results. Their questions on hyperparameter sensitivity and cold-start efficiency were addressed with new empirical data, suggesting a final stance above the acceptance threshold.

---

### Decision · Program_Chairs · 2026-01-26

Accept (Poster)